# WEIGHTED TRAINING FOR CROSS-TASK LEARNING

**Shuxiao Chen**
University of Pennsylvania
shuxiaoc@wharton.upenn.edu

**Koby Crammer**
The Technion
koby@ee.technion.ac.il

**Hangfeng He**
University of Pennsylvania
hangfeng@seas.upenn.edu

**Dan Roth**
University of Pennsylvania
danroth@seas.upenn.edu

**Weijie J. Su**
University of Pennsylvania
suw@wharton.upenn.edu

## ABSTRACT

In this paper, we introduce **T**arget-**A**ware **W**eighted **T**raining (TAWT), a weighted training algorithm for cross-task learning based on minimizing a representation-based task distance between the source and target tasks. We show that TAWT is easy to implement, is computationally efficient, requires little hyperparameter tuning, and enjoys non-asymptotic learning-theoretic guarantees. The effectiveness of TAWT is corroborated through extensive experiments with BERT on four sequence tagging tasks in natural language processing (NLP), including part-of-speech (PoS) tagging, chunking, predicate detection, and named entity recognition (NER). As a byproduct, the proposed representation-based task distance allows one to reason in a theoretically principled way about several critical aspects of cross-task learning, such as the choice of the source data and the impact of fine-tuning.[1]

## 1 INTRODUCTION

The state-of-the-art (SOTA) models in real-world applications rely increasingly on the usage of weak supervision signals (Pennington et al., 2014; Devlin et al., 2019; Liu et al., 2019). Among these, *cross-task* signals are one of the most widely-used weak signals (Zamir et al., 2018; McCann et al., 2018). Despite their popularity, the benefits of cross-task signals are not well understood from a theoretical point of view, especially in the context of deep learning (He et al., 2021; Neyshabur et al., 2020), hence impeding the efficient usage of those signals. Previous work has adopted representation learning as a framework to understand the benefits of cross-task signals, where knowledge transfer is achieved by learning a representation shared across different tasks (Baxter, 2000; Maurer et al., 2016; Tripuraneni et al., 2020; 2021; Du et al., 2021). However, the existence of a shared representation is often too strong an assumption in practice. Such an assumption also makes it difficult to reason about several critical aspects of cross-task learning, such as the quantification of the value of the source data and the impact of fine-tuning (Kalan & Fabian, 2020; Chua et al., 2021).

In this paper, we propose **T**arget-**A**ware **W**eighted **T**raining (TAWT), a weighted training algorithm for efficient cross-task learning. The algorithm can be easily applied to existing cross-task learning paradigms, such as pre-training and joint training, to boost their *sample efficiency* by assigning adaptive (i.e., trainable) weights on the source tasks or source samples. The weights are determined in a theoretically principled way by minimizing a representation-based task distance between the source and target tasks. Such a strategy is in sharp contrast to other weighting schemes common in machine learning, such as importance sampling in domain adaptation (Shimodaira, 2000; Cortes et al., 2010; Jiang & Zhai, 2007).

---

[1] Our code is publicly available at http://cogcomp.org/page/publication_view/963.

The effectiveness of `TAWT` is verified via both theoretical analyses and empirical experiments. Using empirical process theory, we prove a non-asymptotic generalization bound for `TAWT`. The bound is a superposition of two vanishing terms and a term depending on the task distance, the latter of which is potentially negligible due to the re-weighting operation. We then conduct comprehensive experiments on four sequence tagging tasks in NLP: part-of-speech (PoS) tagging, chunking, predicate detection, and named entity recognition (NER). We demonstrate that `TAWT` further improves the performance of BERT (Devlin et al., 2019) in both pre-training and joint training for cross-task learning with limited target data, achieving an average absolute improvement of 3.1% on the performance.

As a byproduct, we propose a representation-based task distance that depends on the quality of representations for each task, respectively, instead of assuming the existence of a single shared representation among all tasks. This finer-grained notion of task distance enables a better understanding of cross-task signals. For example, the representation-based task distance gives an interpretable measure of the value of the source data on the target task based on the discrepancy between their optimal representations. Such a measure is more informative than measuring the difference between tasks via the discrepancy of their task-specific functions (e.g. linear functions) as done in previous theoretical frameworks (Tripuraneni et al., 2020). Furthermore, the representation-based task distance clearly conveys the necessity of fine-tuning: if this distance is non-zero, then fine-tuning the representation becomes necessary as the representation learned from the source data does not converge to the optimal target representation.

Finally, we compare our work with some recent attempts in similar directions. Liu et al. (2020) analyze the benefits of transfer learning by distinguishing source-specific features and transferable features in the source data. Based on the two types of features, they further propose a meta representation learning algorithm to encourage learning transferable and generalizable features. Instead of focusing on the distinction between two types of features, our algorithm and analyses are based on the representation-based task distance and are thus different. Chua et al. (2021) present a theoretical framework for analyzing representations derived from model agnostic meta-learning (Finn et al., 2017), assuming all the tasks use approximately the same underlying representation. In contrast, we do not impose any a priori assumption on the proximity among source and target representations, and our algorithm seeks for a weighting scheme to maximize the proximity. Our work is also different from task weighting in curriculum learning. This line of work tends to learn suitable weights in the stochastic policy to decide which task to study next in curriculum learning (Graves et al., 2017), while `TAWT` aims to learn better representations by assigning more suitable weights on source tasks. Compared to heuristic weighting strategies in multi-task learning (Gong et al., 2019; Zhang & Yang, 2021), we aim to design a practical algorithm with theoretical guarantees for cross-task learning.

## 2   TAWT: TARGET-AWARE WEIGHTED TRAINING

### 2.1   PRELIMINARIES

Suppose we have $T$ source tasks, represented by a collection of probability distributions $\{\mathcal{D}_t\}_{t=1}^T$ on the sample space $\mathcal{X} \times \mathcal{Y}$, where $\mathcal{X} \subseteq \mathbb{R}^d$ is the feature space and $\mathcal{Y} \subseteq \mathbb{R}$ is the label space. For classification problems, we take $\mathcal{Y}$ to be a finite subset of $\mathbb{R}$. We have a single target task, whose probability distribution is denoted as $\mathcal{D}_0$. For the $t$-th task, where $t = 0, 1, \ldots, T$, we observe $n_t$ i.i.d. samples $S_t = \{(\boldsymbol{x}_{ti}, y_{ti})\}_{i=1}^{n_t}$ from $\mathcal{D}_t$. Typically, the number of samples from the target task, $n_0$, is much smaller than the samples from the source tasks, and the goal is to use samples from source tasks to aid the learning of the target task.

Let $\Phi$ be a collection of *representations* from the feature space $\mathcal{X}$ to some latent space $\mathcal{Z} \subseteq \mathbb{R}^r$. We refer to $\Phi$ as the *representation class*. Let $\mathcal{F}$ be a collection of *task-specific functions* from the latent space $\mathcal{Z}$ to the label space $\mathcal{Y}$. The complexity of the representation class $\Phi$ is usually much larger (i.e., more expressive) than that of the task-specific function class $\mathcal{F}$.

Given a bounded loss function $\ell : \mathcal{Y} \times \mathcal{Y} \to [0, 1]$, the optimal pair of representation and task-specific function of the $t$-th task is given by

$$(\phi_t^\star, f_t^\star) \in \operatorname*{argmin}_{\phi_t \in \Phi, f_t \in \mathcal{F}} \mathcal{L}_t(\phi_t, f_t), \qquad \mathcal{L}_t(\phi_t, f_t) := \mathbb{E}_{(X,Y) \sim \mathcal{D}_t}[\ell(f_t \circ \phi_t(X), Y)]. \qquad (2.1)$$

Note that in general, the optimal representations of different tasks are different. For brevity, all proofs for the theory part are deferred to the Appx. A.

## 2.2 DERIVATION OF TAWT

Under the assumption that the optimal representations $\{\phi_t^\star\}_{t=0}^T$ are similar, a representation learned using samples only from the source tasks would perform reasonably well on the target task. Consequently, we can devote $n_0$ samples from the target task to learn only the task specific function. This is a much easier task, since the complexity of $\mathcal{F}$ is typically much smaller than that of $\Phi$.

This discussion leads to a simple yet immensely popular two-step procedure as follows (Tripuraneni et al., 2020; Du et al., 2021). First, we solve a weighted empirical risk minimization problem with respect to the source tasks:

$$(\widehat{\phi}, \{\widehat{f}_t\}_{t=1}^T) \in \operatorname*{argmin}_{\phi \in \Phi, \{f_t\} \subset \mathcal{F}} \sum_{t=1}^T \omega_t \widehat{\mathcal{L}}_t(\phi, f_t), \qquad \widehat{\mathcal{L}}_t(\phi, f_t) := \frac{1}{n_t} \sum_{i=1}^{n_t} \ell(f_t \circ \phi(\boldsymbol{x}_{ti}), y_{ti}), \quad (2.2)$$

where $\boldsymbol{\omega} \in \Delta^{T-1}$ is a user-specified vector lying in the $T$-dimensional probability simplex (i.e., $\sum_{t=1}^T \omega_t = 1$ and $\omega_t \geq 0, \forall 1 \leq t \leq T$). In the second stage, we freeze the representation $\widehat{\phi}$, and seek the task-specific function that minimizes the empirical risk with respect to the target task:

$$\widehat{f}_0 \in \operatorname*{argmin}_{f_0 \in \mathcal{F}} \widehat{\mathcal{L}}_0(\widehat{\phi}, f_0). \tag{2.3}$$

In practice, we can allow $\widehat{\phi}$ to slightly vary (e.g., via fine-tuning) to get a performance boost. In the two-step procedure (2.2)–(2.3), the weight vector $\boldsymbol{\omega}$ is usually taken to be a hyperparameter and is *fixed* during training. Popular choices include the uniform weights (i.e., $\omega_t = 1/T$) or weights proportional to the sample sizes (i.e., $\omega_t = n_t / \sum_{t'=1}^T n_{t'}$) (Liu et al., 2019; Johnson & Khoshgoftaar, 2019). This reveals the *target-agnostic* nature of the two-step procedure (2.2)–(2.3): the weights stay the same regardless the level of proximity between the source tasks and the target task.

Consider the following thought experiment: if we know a priori that the first source task $\mathcal{D}_1$ is closer (compared to other source tasks) to the target task $\mathcal{D}_0$, then we would expect a better performance by raising the importance of $\mathcal{D}_1$, i.e., make $\omega_1$ larger. This thought experiment motivates a *target-aware* procedure that adaptively adjusts the weights based on the proximity of source tasks to the target. A natural attempt for developing such a task-aware procedure is a follows:

$$\min_{\substack{\phi \in \Phi, f_0 \in \mathcal{F}, \\ \boldsymbol{\omega} \in \Delta^{T-1}}} \widehat{\mathcal{L}}_0(\phi, f_0) \qquad \text{subject to } \phi \in \operatorname*{argmin}_{\psi \in \Phi} \min_{\{f_t\} \subset \mathcal{F}} \sum_{t=1}^T \omega_t \widehat{\mathcal{L}}_t(\psi, f_t). \tag{OPT1}$$

That is, we seek for the best weights $\boldsymbol{\omega}$ such that solving (2.2) with this choice of $\boldsymbol{\omega}$ would lead to the lowest training error when we subsequently solve (2.3).

Despite its conceptual simplicity, the formulation (OPT1) is a complicated constrained optimization problem. Nevertheless, we demonstrate that it is possible to transform it into an unconstrained form for which a customized gradient-based optimizer could be applied. To do so, we let $(\phi^{\boldsymbol{\omega}}, \{f_t^{\boldsymbol{\omega}}\})$ be any representation and task-specific functions that minimizes $\sum_{t=1}^T \omega_t \widehat{\mathcal{L}}_t(\phi, f_t)$ over $\phi \in \Phi$ and $\{f_t\} \subset \mathcal{F}$. Equivalently, $\phi^{\boldsymbol{\omega}}$ minimizes $\sum_{t=1}^T \omega_t \min_{f_t \in \mathcal{F}} \widehat{\mathcal{L}}_t(\phi, f_t)$ over $\phi \in \Phi$. With such notations, we can re-write (OPT1) as

$$\min_{f_0 \in \mathcal{F}, \boldsymbol{\omega} \in \Delta^{T-1}} \widehat{\mathcal{L}}_0(\phi^{\boldsymbol{\omega}}, f_0). \tag{OPT2}$$

The gradient of the above objective with respect to the task-specific function, $\nabla_f \widehat{\mathcal{L}}_0(\phi^{\boldsymbol{\omega}}, f_0)$, is easy to calculate via back-propagation. The calculation of the gradient with respect to the weights requires more work, as $\phi^{\boldsymbol{\omega}}$ is an implicit function of $\boldsymbol{\omega}$. By the chain rule, we have $\frac{\partial}{\partial \omega_t} \widehat{\mathcal{L}}_0(\phi^{\boldsymbol{\omega}}, f_0) = [\nabla_\phi \widehat{\mathcal{L}}_0(\phi^{\boldsymbol{\omega}}, f_0)]^\top \frac{\partial}{\partial \omega_t} \phi^{\boldsymbol{\omega}}$. Since $\phi^{\boldsymbol{\omega}}$ is a minimizer of $\phi \mapsto \sum_{t=1}^T \omega_t \min_{f_t \in \mathcal{F}} \widehat{\mathcal{L}}_t(\phi, f_t)$, we have

$$F(\phi^{\boldsymbol{\omega}}, \boldsymbol{\omega}) = 0, \ \forall \boldsymbol{\omega} \in \Delta^{T-1}, \qquad F(\phi, \boldsymbol{\omega}) := \nabla_\phi \sum_{t=1}^T \omega_t \min_{f_t \in \mathcal{F}} \widehat{\mathcal{L}}_t(\phi, f_t). \tag{2.4}$$

By implicit function theorem, if $F(\cdot, \cdot)$ is everywhere differentiable and the matrix $\partial F(\phi, \boldsymbol{\omega})/\partial \phi$ is invertible for any $(\phi, \boldsymbol{\omega})$ near some $(\widetilde{\phi}, \widetilde{\boldsymbol{\omega}})$ satisfying $F(\widetilde{\phi}, \widetilde{\boldsymbol{\omega}}) = 0$, then we can conclude that the

---

**Algorithm 1:** **T**arget-**A**ware **W**eighted **T**raining (`TAWT`)

---

**Input:** Datasets $\{S_t\}_{t=0}^T$.
**Output:** Final pair of representation and task-specific function $(\widehat{\phi}, \widehat{f}_0)$ for the target task.
Initialize parameters $\boldsymbol{\omega}^0 \in \Delta^{T-1}, \phi^0 \in \Phi, \{f_t^0\}_{t=0}^T \subset \mathcal{F}$;
**for** $k = 0, \ldots, K-1$ **do**

    Starting from $(\phi^k, \{f_t^k\}_{t=1}^T)$, run a few steps of SGD to get $(\phi^{k+1}, \{f_t^{k+1}\}_{t=1}^T)$;

    Use the approximate gradient $\nabla_f \widehat{\mathcal{L}}_0(\phi^{k+1}, f_0)$ to run a few steps of SGD from $f_0^k$ to get $f_0^{k+1}$;

    Run one step of approximate mirror descent (2.7)–(2.8) from $\boldsymbol{\omega}^k$ to get $\boldsymbol{\omega}^{k+1}$;

**end**
**return** $\widehat{\phi} = \phi^K, \widehat{f}_0 = f_0^K$

---

map $\boldsymbol{\omega} \mapsto \phi^{\boldsymbol{\omega}}$ is a locally well-defined function near $\widetilde{\boldsymbol{\omega}}$, and the derivative of this map is given by

$$\frac{\partial}{\partial \omega_t} \phi^{\boldsymbol{\omega}} = -\left( \frac{\partial F(\phi, \boldsymbol{\omega})}{\partial \phi} \Big|_{\phi = \phi^{\boldsymbol{\omega}}} \right)^{-1} \left( \frac{\partial F(\phi, \boldsymbol{\omega})}{\partial \omega_t} \Big|_{\phi = \phi^{\boldsymbol{\omega}}} \right). \tag{2.5}$$

To simplify the above expression, note that under regularity conditions, we can regard $\nabla_\phi \widehat{\mathcal{L}}_t(\phi, f_t^{\boldsymbol{\omega}})$ as a sub-gradient of the map $\phi \mapsto \min_{f_t \in \mathcal{F}} \widehat{\mathcal{L}}_t(\phi, f_t)$. This means that we can write $F(\phi^{\boldsymbol{\omega}}, \boldsymbol{\omega}) = \sum_{t=1}^T \omega_t \nabla_\phi \widehat{\mathcal{L}}_t(\phi^{\boldsymbol{\omega}}, f_t^{\boldsymbol{\omega}})$. Plugging this expression back to (2.5) and recalling the expression for $\partial \widehat{\mathcal{L}}_0(\phi^{\boldsymbol{\omega}}, f_0)/\partial \omega_t$ derived via the chain rule, we get

$$\frac{\partial}{\partial \omega_t} \widehat{\mathcal{L}}_0(\phi^{\boldsymbol{\omega}}, f_0) = -[\nabla_\phi \widehat{\mathcal{L}}_0(\phi^{\boldsymbol{\omega}}, f_0)]^\top \Big[ \sum_{t=1}^T \omega_t \nabla_\phi^2 \widehat{\mathcal{L}}_t(\phi^{\boldsymbol{\omega}}, f_t^{\boldsymbol{\omega}}) \Big]^{-1} [\nabla_\phi \widehat{\mathcal{L}}_t(\phi^{\boldsymbol{\omega}}, f_t^{\boldsymbol{\omega}})]. \tag{2.6}$$

Now that we have the expressions for the gradients of $\widehat{\mathcal{L}}_0(\phi^{\boldsymbol{\omega}}, f_0)$ with respect to $f_0$ and $\boldsymbol{\omega}$, we can solve (OPT2) via a combination of alternating minimization and mirror descent. To be more specific, suppose that at iteration $k$, the current weights, representation, and task-specific functions are $\boldsymbol{\omega}^k, \phi^k$, and $\{f_t^k\}_{t=0}^T$, respectively. At this iteration, we conduct the following three steps:

1. Freeze $\boldsymbol{\omega}^k$. Starting from $(\phi^k, \{f_t^k\}_{t=1}^T)$, run a few steps of SGD on the objective function $(\phi, \{f_t\}_{t=1}^T) \mapsto \sum_{t=1}^T \omega_t^k \widehat{\mathcal{L}}_t(\phi, f_t)$ to get $(\phi^{k+1}, \{f_t^{k+1}\}_{t=1}^T)$, which is regarded as an approximation of $(\phi^{\boldsymbol{\omega}^k}, \{f_t^{\boldsymbol{\omega}^k}\}_{t=1}^T)$;

2. Freeze $(\phi^{k+1}, \{f_t^{k+1}\}_{t=1}^T)$. Approximate the gradient $\nabla_f \widehat{\mathcal{L}}_0(\phi^{\boldsymbol{\omega}^k}, f_0)$ by $\nabla_f \widehat{\mathcal{L}}_0(\phi^{k+1}, f_0)$. Using this approximate gradient, run a few steps of SGD from $f_0^k$ to get $f_0^{k+1}$;

3. Freeze $(\phi^{k+1}, \{f_t^{k+1}\}_{t=0}^T)$. Approximate the partial derivative $\partial \widehat{\mathcal{L}}_0(\phi^{\boldsymbol{\omega}^k}, f_0^{k+1})/\partial \omega_t$ by

$$g_t^k := -[\nabla_\phi \widehat{\mathcal{L}}_0(\phi^{k+1}, f_0^{k+1})]^\top \Big[ \sum_{t=1}^T \omega_t \nabla_\phi^2 \widehat{\mathcal{L}}_t(\phi^{k+1}, f_t^{k+1}) \Big]^{-1} [\nabla_\phi \widehat{\mathcal{L}}_t(\phi^{k+1}, f_t^{k+1})]. \tag{2.7}$$

Then run one step of mirror descent (with step size $\eta^k$) from $\boldsymbol{\omega}^k$ to get $\boldsymbol{\omega}^{k+1}$:

$$\omega_t^{k+1} = \frac{\omega_t^k \exp\{-\eta^k g_t^k\}}{\sum_{t'=1}^T \omega_{t'}^k \exp\{-\eta^k g_{t'}^k\}}. \tag{2.8}$$

We use mirror descent in (2.8), as it is a canonical generalization of Euclidean gradient descent to gradient descent on the probability simplex (Beck & Teboulle, 2003). Note that other optimization methods, such as projected gradient descent, can also be used here. The update rule (2.8) has a rather intuitive explanation. Note that $g_t^k$ is a weighted dissimilarity measure between the gradients $\nabla_\phi \widehat{\mathcal{L}}_0$ and $\nabla_\phi \widehat{\mathcal{L}}_t$. This can further be regarded as a crude dissimilarity measure between the optimal representations of the target task and the $t$-th source task. The mirror descent updates $\omega_t$ along the direction where the target task and the $t$-th source task are more similar. The overall procedure is summarized in Algorithm 1.

A faithful implementation of the above steps would require a costly evaluation of the inverse of the Hessian matrix $\sum_{t=1}^T \omega_t \nabla_\phi^2 \widehat{\mathcal{L}}_t(\phi^{k+1}, f_t^{k+1}) \in \mathbb{R}^{r \times r}$. In practice, we can bypass this step by

replacing[2] the Hessian-inverse-weighted dissimilarity measure (2.7) with a consine-similarity-based dissimilarity measure (see Section 4 for details).

The previous derivation has focused on weighted pre-training, i.e., the target data is not used when defining the constrained set in (OPT1). It can be modified, *mutatis mutandis*, to handle weighted joint-training, where we change (OPT1) to

$$\min_{\substack{\phi \in \Phi, f_0 \in \mathcal{F}, \\ \boldsymbol{\omega} \in \Delta^T}} \widehat{\mathcal{L}}_0(\phi, f_0) \qquad \text{subject to } \phi \in \operatorname*{argmin}_{\psi \in \Phi} \min_{\{f_t\} \subset \mathcal{F}} \sum_{t=0}^{T} \omega_t \widehat{\mathcal{L}}_t(\psi, f_t). \tag{2.9}$$

Compared to (OPT1), we now also use the data from the target task when learning the representation $\phi$, and thus there is an extra weight $\omega_0$ on the target task. The algorithm can also be easily extended to handle multiple target tasks or to put weights on samples (as opposed to putting weights on tasks). The algorithm could also be applied to improve the efficiency of learning from cross-domain and cross-lingual signals, and we postpone such explorations for future work.

## 3 THEORETICAL GUARANTEES

### 3.1 A REPRESENTATION-BASED TASK DISTANCE

In this subsection, we introduce a representation-based task distance, which will be crucial in the theoretical understanding of `TAWT`. To start with, let us define the representation and task-specific functions that are optimal in an "$\boldsymbol{\omega}$-weighted sense" as follows:

$$(\bar{\phi}^{\boldsymbol{\omega}}, \{\bar{f}_t^{\boldsymbol{\omega}}\}_{t=1}^T) \in \operatorname*{argmin}_{\phi \in \Phi, \{f_t\} \subset \mathcal{F}} \sum_{t=1}^{T} \omega_t \mathcal{L}_t(\phi, f_t) \tag{3.1}$$

Intuitively, $(\bar{\phi}^{\boldsymbol{\omega}}, \{\bar{f}_t^{\boldsymbol{\omega}}\})$ are optimal on the $\boldsymbol{\omega}$-weighted source tasks when only a single representation is used. Since $\bar{\phi}^{\boldsymbol{\omega}}$ may not be unique, we introduce the function space $\bar{\Phi}^{\boldsymbol{\omega}} \subset \Phi$ to collect all $\bar{\phi}^{\boldsymbol{\omega}}$s that satisfy (3.1). To further simplify the notation, we let $\mathcal{L}_t^{\star}(\phi) := \min_{f_t \in \mathcal{F}} \mathcal{L}_t(\phi, f_t)$, which stands for the risk incurred by the representation $\phi$ on the $t$-th task. With the foregoing notation, we can write $\bar{\phi}^{\boldsymbol{\omega}} \in \operatorname{argmin}_{\phi \in \Phi} \sum_{t=1}^{T} \omega_t \mathcal{L}_t^{\star}(\phi)$. The definition of the task distance is given below.

**Definition 3.1** (Representation-based task distance). *The representation-based task distance between the $\boldsymbol{\omega}$-weighted source tasks and the target task is defined as*

$$\texttt{dist}\Big(\sum_{t=1}^{T} \omega_t \mathcal{D}_t, \mathcal{D}_0\Big) := \sup_{\bar{\phi}^{\boldsymbol{\omega}} \in \bar{\Phi}^{\boldsymbol{\omega}}} \mathcal{L}_0^{\star}(\bar{\phi}^{\boldsymbol{\omega}}) - \mathcal{L}_0^{\star}(\phi_0^{\star}), \tag{3.2}$$

*where the supremum is taken over any $\bar{\phi}^{\boldsymbol{\omega}}$ satisfying (3.1), and $\phi_0^{\star}$ is the optimal target representation.*

If all the tasks share the same optimal representation, then above distance becomes exactly zero. Under such an assumption, the only source of discrepancy among tasks arises from the difference in their task-specific functions. This can be problematic in practice, as the task-specific functions alone are usually not expressive enough to describe the intrinsic difference among tasks. In contrast, we relax the shared representation assumption substantially by allowing the optimal representations to differ and the distance to be non-zero.

The above notion of task distance also allows us to reason about certain important aspects of cross-task learning. For example, this task distance is asymmetric, capturing the asymmetric nature of cross-task learning. Moreover, if the task distance is non-zero, then fine-tuning the representation becomes necessary, because solving (2.2) alone would gives a representation that does not converge to the correct target $\phi_0^{\star}$. In addition, an empirical illustration of the representation-based task distance can be found in Fig. 3 in Appx. B.

This task distance can be naturally estimated from the data by replacing all population quantities with their empirical version. Indeed, minimizing the estimated task distance over the weights is equivalent to the optimization formulation (OPT1) of our algorithm (see Appx. A.1 for a detailed derivation). This observation gives an alternative and *theoretically principled derivation* of `TAWT`.

---

[2] This type of approximation is common and almost necessary, such as MAML (Finn et al., 2017).

## 3.2 PERFORMANCE GUARANTEES FOR TAWT

To give theoretical guarantees for TAWT, we need a few standard technical assumptions. The first one concerns the Lipschitzness of the function classes, and the second one controls the complexity of the function classes via uniform entropy (Wellner & van der Vaart, 2013) as follows.

**Assumption A** (Lipschitzness). *The loss function $\ell : \mathcal{Y} \times \mathcal{Y} \to [0,1]$ is $L_\ell$-Lipschitz in the first argument, uniformly over the second argument. Any $f \in \mathcal{F}$ is $L_\mathcal{F}$-Lipschitz w.r.t. the $\ell_2$ norm.*

**Assumption B** (Uniform entropy control of function classes). *There exist $C_\Phi > 0, \nu_\Phi > 0$, such that for any probability measure $\mathbb{Q}_\mathcal{X}$ on $\mathcal{X} \subseteq \mathbb{R}^d$, we have*

$$\mathcal{N}(\Phi; L^2(\mathbb{Q}_\mathcal{X}); \varepsilon) \leq (C_\Phi/\varepsilon)^{\nu_\Phi}, \qquad \forall \varepsilon > 0, \tag{3.3}$$

*where $\mathcal{N}(\Phi; L^2(\mathbb{Q}_\mathcal{X}); \varepsilon)$ is the $L^2(\mathbb{Q}_\mathcal{X})$ covering number of $\Phi$ (i.e., the minimum number of $L^2(\mathbb{Q}_\mathcal{X})$ balls[3] with radius $\varepsilon$ required to cover $\Phi$). In parallel, there exist $C_\mathcal{F} > 0, \nu_\mathcal{F} > 0$, such that for any probability measure $\mathbb{Q}_\mathcal{Z}$ on $\mathcal{Z} \subseteq \mathbb{R}^r$, we have*

$$\mathcal{N}(\mathcal{F}; L^2(\mathbb{Q}_\mathcal{Z}); \varepsilon) \leq (C_\mathcal{F}/\varepsilon)^{\nu_\mathcal{G}}, \qquad \forall \varepsilon > 0, \tag{3.4}$$

*where $\mathcal{N}(\mathcal{F}; L^2(\mathbb{Q}_\mathcal{Z}); \varepsilon)$ is the $L^2(\mathbb{Q}_\mathcal{Z})$ covering number of $\mathcal{F}$.*

Uniform entropy generalizes the notion of Vapnik-Chervonenkis dimension (Vapnik, 2013) and allows us to give a unified treatment of regression and classification problems. For this reason, function classes satisfying the above assumption are also referred to as "VC-type classes" in the literature (Koltchinskii, 2006). In particular, if each coordinate of $\Phi$ has VC-subgraph dimension $c(\Phi)$, then (3.3) is satisfied with $\nu_\Phi = \Theta(r \cdot c(\Phi))$ (recall that $r$ is the dimension of the latent space $\mathcal{Z}$). Similarly, if $\mathcal{F}$ has VC-subgraph dimension $c(\mathcal{F})$, then (3.4) is satisfied with $\nu_\mathcal{F} = \Theta(c(\mathcal{F}))$.

The following definition characterizes how "transferable" a representation $\phi$ is from the $\boldsymbol{\omega}$-weighted source tasks to the target task.

**Definition 3.2** (Transferability). *A representation $\phi \in \Phi$ is $(\rho, C_\rho)$-transferable from $\boldsymbol{\omega}$-weighted source tasks to the target task, if there exists $\rho > 0, C_\rho > 0$ such that for any $\bar{\phi}^{\boldsymbol{\omega}} \in \bar{\Phi}^{\boldsymbol{\omega}}$, we have*

$$\mathcal{L}_0^\star(\phi) - \mathcal{L}_0^\star(\bar{\phi}^{\boldsymbol{\omega}}) \leq C_\rho \left( \sum_{t=1}^T \omega_t [\mathcal{L}_t^\star(\phi) - \mathcal{L}_t^\star(\bar{\phi}^{\boldsymbol{\omega}})] \right)^{1/\rho}. \tag{3.5}$$

Intuitively, the above definition says that relative to $\bar{\phi}^{\boldsymbol{\omega}}$, the risk of $\phi$ on the target task can be controlled by a polynomial of the average risk of $\phi$ on the source tasks. This can be regarded as an adaptation of the notions of transfer exponent and relative signal exponent (Hanneke & Kpotufe, 2019; Cai & Wei, 2021) originated from the transfer learning literature to the representation learning setting. This can also be seen as a generalization of the task diversity assumption (Tripuraneni et al., 2020; Du et al., 2021) to the case when the optimal representations $\{\phi_t^\star\}_{t=1}^T$ do not coincide. In addition, Tripuraneni et al. (2020) and Du et al. (2021) proved that $\rho = 1$ in certain simple models and under some simplified setting.

In this part, we prove a non-asymptotic generalization bound for TAWT. The fact that the weights are learned from the data substantially complicates the analysis. To proceed further, we make a few simplifying assumptions. First, we assume that the sample sizes of the source data are relatively balanced: there exists an integer $n$ such that $n_t = \Theta(n)$ for any $t \in \{1, \ldots, T\}$. Meanwhile, instead of directly analyzing (OPT1), we focus on its sample split version. In particular, we let $B_1 \cup B_2$ be a partition of $\{1, \ldots, n_0\}$, where $|B_1| = \Theta(|B_2|)$. Define

$$\widehat{\mathcal{L}}_0^{(1)}(\phi, f_0) := \frac{1}{|B_1|} \sum_{i \in B_1} \ell(f_0 \circ \phi(\boldsymbol{x}_{0i}), y_{0i}), \qquad \widehat{\mathcal{L}}_0^{(2)}(\phi, f_0) := \frac{1}{|B_2|} \sum_{i \in B_2} \ell(f_0 \circ \phi(\boldsymbol{x}_{0i}), y_{0i}).$$

We first solve (OPT1) restricted to the first part of the data:

$$(\widehat{\phi}, \widehat{\boldsymbol{\omega}}) \in \operatorname*{argmin}_{\phi \in \Phi, \boldsymbol{\omega} \in \Delta^{T-1}} \min_{f_0 \in \mathcal{F}} \widehat{\mathcal{L}}_0^{(1)}(\phi, f_0) \qquad \text{subject to } \phi \in \operatorname*{argmin}_{\psi \in \Phi} \min_{\{f_t\} \subset \mathcal{F}} \sum_{t=1}^T \omega_t \widehat{\mathcal{L}}_t(\psi, f_t). \tag{3.6}$$

---

[3]For two vector-valued functions $\phi, \psi \in \Phi$, their $L^2(\mathbb{Q}_\mathcal{X})$ distance is $\left( \int \|\phi(\boldsymbol{x}) - \psi(\boldsymbol{x})\|^2 d\mathbb{Q}_\mathcal{X}(\boldsymbol{x}) \right)^{1/2}$.

Then, we proceed by solving

$$\widehat{f}_0 \in \underset{f_0 \in \mathcal{F}}{\operatorname{argmin}} \, \widehat{\mathcal{L}}_0^{(2)}(\widehat{\phi}, f_0). \tag{3.7}$$

Such a sample splitting ensures the independence of $\widehat{\phi}$ and the second part of target data $B_2$, hence allowing for more transparent theoretical analyses. Such strategies are common in statistics and econometrics literature when the algorithm has delicate dependence structures (see, e.g., (Hansen, 2000; Chernozhukov et al., 2018)). We emphasize that sample splitting is conducted only for theoretical convenience and is not used in practice.

The following theorem gives performance guarantees for the sample split version of TAWT.

**Theorem 3.1** (Performance of TAWT with sample splitting). *Let $(\widehat{\phi}, \widehat{f}_0)$ be obtained via solving (3.6)–(3.7). Let Assumptions A and B hold. In addition, assume that the learned weights satisfy $\widehat{\boldsymbol{\omega}} \in \mathcal{W}_\beta := \{\boldsymbol{\omega} \in \Delta^{T-1} : \beta^{-1} \leq \omega_t/\omega_{t'} \leq \beta, \forall t \neq t'\}$, where $\beta \geq 1$ is an absolute constant. Fix $\delta \in (0, 1)$. There exists a constant $C = C(L_\ell, L_\mathcal{F}, C_\Phi, C_\mathcal{F}) > 0$ such that the following holds: if for any weights $\boldsymbol{\omega} \in \mathcal{W}_\beta$ and any representation $\phi$ in a $C\beta \cdot \sqrt{(\nu_\Phi \log \delta^{-1})/nT + (\nu_\mathcal{F} + \log T)/n}$-neighborhood of $\bar{\Phi}^{\boldsymbol{\omega}}$[4], there exists a specific $\bar{\phi}^{\boldsymbol{\omega}} \in \bar{\Phi}^{\boldsymbol{\omega}}$ such that $\bar{\phi}^{\boldsymbol{\omega}}$ is $(\rho, C_\rho)$-transferable, then there exists another $C' = C'(L_\ell, L_\mathcal{F}, C_\Phi, C_\mathcal{F}, C_\rho, \rho)$ such that with probability at least $1 - \delta$, we have*

$$\mathcal{L}_0(\widehat{\phi}, \widehat{f}_0) - \mathcal{L}_0(\phi_0^\star, f_0^\star) \leq C'\left[\left(\frac{\nu_\mathcal{F} + \log(1/\delta)}{n_0}\right)^{\frac{1}{2}} + \beta^{1/\rho}\left(\frac{\nu_\Phi + \log(1/\delta)}{nT} + \frac{\nu_\mathcal{F} + \log T}{n}\right)^{\frac{1}{2\rho}}\right]$$
$$+ \texttt{dist}\left(\sum_{t=1}^T \widehat{\omega}_t \mathcal{D}_t, \mathcal{D}_0\right). \tag{3.8}$$

The upper bound in (3.8) is a superposition of three terms. Let us disregard the $\log(1/\delta)$ terms for now and focus on the dependence on the sample sizes and problem dimensions. The first term, which scales with $\sqrt{\nu_\mathcal{F}/n_0}$, corresponds to the error of learning the task-specific function in the target task. This is unavoidable even if the optimal representation $\phi_0^\star$ is known.

The second term that scales with $[(\nu_\Phi + T\nu_\mathcal{F})/(nT)]^{1/2\rho}$ characterizes the error of learning the imperfect representation $\bar{\phi}^{\boldsymbol{\omega}}$ from the source datasets and transferring the knowledge to the target task. Note that this term is typically much smaller than $\sqrt{(\nu_\Phi + \nu_\mathcal{F})/n_0}$, the error that would have been incurred when learning only from the target data, thus illustrating the potential benefits of representation learning. This happens because $nT$ is typically much larger than $n_0$.

The third term is precisely the task distance introduced in Definition 3.1. The form of the task distance immediately demonstrates the possibility of "matching" $\bar{\phi}^{\boldsymbol{\omega}} \approx \phi_0^\star$ via varying the weights $\boldsymbol{\omega}$, under which case the third term would be negligible compared to the former two terms. For example, in Appx. A.2, we give a sufficient condition for exactly matching $\bar{\phi}^{\boldsymbol{\omega}} = \phi_0^\star$.

The proof of Theorem 3.1 is based on empirical process theory. Along the way, we also establish an interesting result on multi-task learning, where the goal is to improve the average performance for all tasks instead of target tasks (see Lemma A.1 in Appx. A). The current analysis can also be extended to cases where multiple target tasks are present.

## 4 EXPERIMENTS

In this section, we verify the effectiveness of TAWT in extensive experiments using four NLP tasks, including PoS tagging, chunking, predicate detection, and NER. More details are in Appx. B.

**Experimental settings.** In our experiments, we mainly use two widely-used NLP datasets, Ontontes 5.0 (Hovy et al., 2006) and CoNLL-2000 (Tjong Kim Sang & Buchholz, 2000). Ontonotes 5.0 contains annotations for PoS tagging, predicate detection, and NER, and CoNLL-2000 is a shared task for chunking. There are about $116K$ sentences, $16K$ sentences, and $12K$ sentences in the training, development, and test sets for tasks in Ontonotes 5.0. As for CoNLL-2000, there are

---

[4]A representation $\phi$ is in an $\varepsilon$-neighborhood of $\bar{\Phi}^{\boldsymbol{\omega}}$ if $\sum_{t=1}^T \omega_t[\mathcal{L}_t^\star(\phi) - \mathcal{L}_t^\star(\bar{\phi}^{\boldsymbol{\omega}})] \leq \varepsilon$.

| Target Task / Learning Paradigm | PoS | Chunking | Predicate Detection | NER | Avg |
|---|---|---|---|---|---|
| Single-Task Learning | 34.37 | 43.05 | 66.26 | 33.20 | 44.22 |
| Pre-Training | 49.43 | 73.15 | 74.10 | 41.22 | 59.48 |
| Weighted Pre-Training | **51.17** *** | **73.41** | **75.77** *** | **46.23** *** | **61.64** |
| Joint Training | 53.83 | 75.58 | 75.42 | 43.50 | 62.08 |
| Weighted Joint Training | **57.34** *** | **77.78** *** | **75.98** *** | **53.44** *** | **66.14** |
| Normalized Joint Training | 84.14 | 88.91 | **77.02** | 61.15 | 77.80 |
| Weighted Normalized Joint Training | **86.07** *** | **90.62** *** | 76.67 | **63.44** *** | **79.20** |

Table 1: The benefits of weighted training for different learning paradigms under different settings. There are four tasks in total, PoS tagging, chunking, predicate detection, and NER. For each setting, we choose one task as the target task and the remaining three tasks as source tasks. We randomly choose $9K$ training sentences for each source task respectively, because the training size of the chunking dataset is 8936. As for the target task, we randomly choose 100, 100, 300, 500 training sentences for PoS tagging, chunking, predicate detection, and NER respectively, based on the difficulty of tasks. Single-task learning denotes learning only with the small target data. *** indicates the p-value of the paired sampled t-test is smaller than $0.001$.

about $9K$ sentences and $2K$ sentences in the training and test sets. As for the evaluation metric, we use accuracy for PoS tagging, span-level F1 for chunking, word-level F1 for predicate detection, and span-level F1 for NER. We use BERT[5] (Devlin et al., 2019) as our basic model in our main experiments. Specifically, we use the pre-trained case-sensitive BERT-base PyTorch implementation (Wolf et al., 2020), and the common hyperparameters for BERT. In the BERT, the task-specific function is the last-layer linear classifier, and the representation model is the remaining part. As for cross-task learning paradigms, we consider two popular learning paradigms, pre-training, and joint training. Pre-training first pre-train the representation part on the source data and then fine-tune the whole target model on the target data. Joint training uses both source and target data to train the shared representation model and task-specific functions for both source and target tasks at the same time. As for the multi-task learning part in both pre-training and joint training, we adopt the same multi-task learning algorithm as in MT-DNN (Liu et al., 2019). More explanation on the choice of experimental settings can be found in Appx. C.

**Settings for weighted training.** For scalability, in all the experiments, we approximate $g_t^k$ in Eq. (2.7) by $-c \times \mathtt{sim}(\nabla_\phi \widehat{\mathcal{L}}_0(\phi^{k+1}, f_0^{k+1}), \nabla_\phi \widehat{\mathcal{L}}_t(\phi^{k+1}, f_t^{k+1}))$, where $\mathtt{sim}(\cdot, \cdot)$ denotes the cosine similarity between two vectors. Note that this type of approximation is common and almost necessary, such as MAML (Finn et al., 2017). For weighted joint training, we choose $c = 1$. For weighted pre-training, we choose the best $c$ among $[0.3, 1, 10, 30, 100]$, because the cosine similarity between the pre-training tasks and the target task is small in general. In practice, we further simplify the computation of $\nabla_\phi \widehat{\mathcal{L}}_t(\phi^{k+1}, f_t^{k+1})$ $(t = 0, 1, \ldots, T)$ in Eq. (2.7) by computing the gradient over the average loss of a randomly sampled subset of the training set instead of the average loss of the whole training set as in Eq. (2.2). In our experiments, we simply set the size of the randomly sampled subset of the training set as $64$, though a larger size is more beneficial in general. In our experiments, we choose $\eta^k = 1.0$ in the mirror descent update (2.8). It is worthwhile to note that there is no need to tune any extra hyperparameters for weighted joint training, though we believe that the performance of TAWT can be further improved by tuning extra hyper parameters, such as the learning rate $\eta^k$.

**Results.** The effectiveness of TAWT is first demonstrated for both pre-training and joint training on four tasks with quite a few training examples in the target data, as shown in Table 1. Experiments with more target training examples and some additional experiments can be found in Table 2 and Table 3 in Appx. B. Finally, we note that TAWT can be easily extended from putting weights on tasks to putting weights on samples (see Fig. 2 in Appx. B).

**Normalized joint training.** Inspired by the final task weights learned by TAWT (see Table 6 in Appx. B), we also experiment with normalized joint training. The difference between joint training and normalized joint training lies in the initialization of task weights. For (weighted) joint training, the weights on tasks are initialized to be $\omega_t = n_t / \sum_{t'=1}^{T} n_{t'}$ (i.e, the weights on the loss of each

---
[5]While BERT is no longer the SOTA model, all SOTA models are slight improvements of BERT, so our experiments are done with highly competitive models.

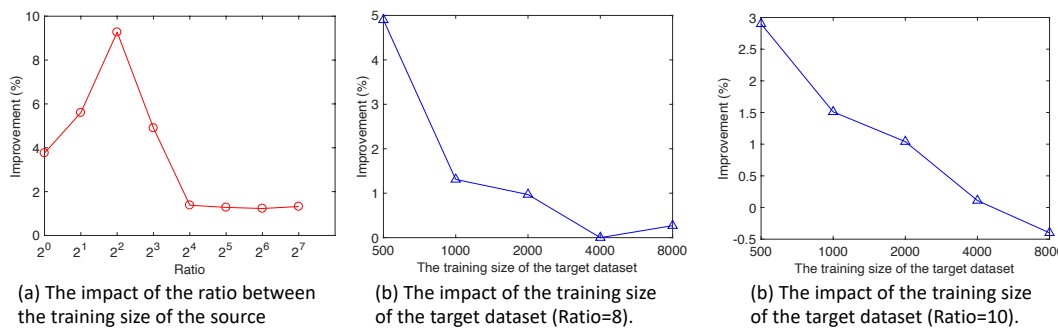

(a) The impact of the ratio between the training size of the source dataset and the target dataset.

(b) The impact of the training size of the target dataset (Ratio=8).

(b) The impact of the training size of the target dataset (Ratio=10).

Figure 1: Analysis of the weighted training algorithms. We analyze the impact of two crucial factors on the improvement of the weighted training algorithms: the ratio between the training sizes of the source and target datasets, and the training size of the target dataset. In this figure, we use NER as the target task and source tasks include PoS tagging and predicate detection. In the first subfigure, we keep the training size of the target tasks as $500$ and change the ratio from $1$ to $128$ on a log scale. In the second (third) subfigure, we keep the ratio as $8$ ($10$) and change the training size of the target dataset from $500$ to $8000$ on a log scale. The corresponding improvement from the normalized joint training to the weighted normalized joint training is shown.

example will be uniform), whereas for the (weighted) normalized joint training, the weights on tasks are initialized to be $\omega_t = 1/T$ (i.e., the weight on the loss of each example will be normalized by the sample sizes). Note that the loss for each task is already normalized by the sample size in our theoretical analysis (see Eq. 2.2). As a byproduct, we find that normalized joint training is much better than the widely used joint training when the source data is much larger than the target data. In addition, TAWT can still be used to further improve normalized joint training. The corresponding results can be found in Table 1. In addition, we find that dynamic weights might be a better choice in weighted training compared to fixed weights (see Table 7 in Appx. B).

**Analysis.** Furthermore, we analyze two crucial factors (i.e., the ratio between the training sizes of the source and target datasets, and the training size of the target dataset) that affect the improvement of TAWT in Fig. 1. In general, we find that TAWT is more beneficial when the performance of the base model is poorer, either because of a smaller-sized target data or due to a smaller ratio between the source data size and the target data size. More details are in Table 4 and Table 5 in Appx. B.

## 5 DISCUSSION

In this paper, we propose a new weighted training algorithm, TAWT, to improve the *sample efficiency* of learning from cross-task signals. TAWT adaptively assigns weights on tasks or samples in the source data to minimize the representation-based task distance between source and target tasks. The algorithm is an easy-to-use plugin that can be applied to existing cross-task learning paradigms, such as pre-training and joint training, without introducing too much computational overhead and hyperparameters tuning. The effectiveness of TAWT is further corroborated through theoretical analyses and empirical experiments. To the best of our knowledge, TAWT is the first weighted algorithm for cross-task learning with theoretical guarantees, and the proposed representation-based task distance also sheds light on many critical aspects of cross-task learning.

**Limitations and Future Work.** There are two main limitations in our work. First, although we gave an efficient implementation of task-weighted version of TAWT, an efficient implementation of sample-weighted version of TAWT is still lacking, and we leave it for future work. Second, we are not aware of any method that can efficiently estimate the representation-based task distance without training the model, and we plan to work more on this direction. In addition, we also plan to evaluate TAWT in other settings, such as cross-domain and cross-lingual settings; in more general cases, such as multiple target tasks in the target data; and in other tasks, such as language modeling, question answering, sentiment analysis, image classification, and object detection.

ACKNOWLEDGMENTS

This material is based upon work supported by the US Defense Advanced Research Projects Agency (DARPA) under contracts FA8750-19-2-0201 and W911NF-20-1-0080, NSF through CAREER DMS-1847415 and an Alfred Sloan Research Fellowship. The views expressed are those of the authors and do not reflect the official policy or position of the Department of Defense or the U.S. Government.

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

## A   OMITTED PROOFS

We start by introducing some notations. For a set $S$, we let $\mathbb{1}_S$ be its indicator function and we use $\#S$ and $|S|$ interchangeably to denote its cardinality. For two positive sequences $\{a_n\}$ and $\{b_n\}$, we write $a_n \lesssim b_n$ or $a_n = \mathcal{O}(b_n)$ to denote $\limsup a_n/b_n < \infty$, and we let $a_n \gtrsim b_n$ or $a_n = \Omega(b_n)$ to denote $b_n \lesssim a_n$. Meanwhile, the notation $a_n \asymp b_n$ or $a_n = \Theta(b_n)$ means $a_n \lesssim b_n$ and $a_n \gtrsim b_n$ simultaneously. For a vector $\boldsymbol{x}$, we let $\|\boldsymbol{x}\|$ denote its $\ell_2$ norm. In this section, we treat $L_\ell, L_\mathcal{F}, C_\Phi, C_\mathcal{F}$ as absolute constants and we hide the dependence on those parameters in our theoretical results. The exact dependence on those parameters can be easily traced from the proofs.

### A.1   TAWT AND TASK DISTANCE MINIMIZATION

If we estimate $\mathcal{L}_0^\star(\phi) = \min_{f_0 \in \mathcal{F}} \mathcal{L}_0(\phi, f_0)$ by $\min_{f_0 \in \mathcal{F}} \widehat{\mathcal{L}}_0(\phi, f_0)$ and $\bar{\phi}^{\boldsymbol{\omega}} \in \operatorname{argmin}_{\phi \in \Phi} \sum_{t=1}^T \omega_t \cdot \min_{f_t \in \mathcal{F}} \mathcal{L}_t(\phi, f_t)$ by the argmin of $\sum_{t=1}^T \omega_t \cdot \min_{f_t \in \mathcal{F}} \widehat{\mathcal{L}}_t(\phi, f_t)$, then overall, the quantity $\mathcal{L}_0^\star(\bar{\phi}^{\boldsymbol{\omega}})$ can be estimated by

$$\min_{f_0 \in \mathcal{F}} \widehat{\mathcal{L}}_0(\phi, f_0) \qquad \text{subject to } \phi \in \operatorname*{argmin}_{\psi \in \Phi} \sum_{t=1}^T \omega_t \widehat{\mathcal{L}}_t(\psi, f_t).$$

Thus, minimizing the estimated task distance over the weights (note that $\mathcal{L}_0^\star(\phi^\star)$ is a constant) is equivalent to the optimization formulation (OPT1) of our algorithm.

To further relate TAWT with task distance minimization, we provided an analysis of the two-step procedure (2.2)–(2.3) with fixed weights. The next theorem gives the corresponding performance guarantees.

**Theorem A.1** (Performance of the two-step procedure with fixed weights). *Let $(\widehat{\phi}, \widehat{f}_0)$ be obtained via solving (2.2)–(2.3) with fixed $\boldsymbol{\omega}$. Let Assumptions A and B hold. Fix any $\delta \in (0,1)$ and define $N_{\boldsymbol{\omega}} = (\sum_{t=1}^T \omega_t^2/n_t)^{-1}$. There exists a constant $C = C(L_\ell, L_\mathcal{F}, C_\Phi, C_\mathcal{F}) > 0$ such that the following holds: if for any representation $\phi$ in a $C\sqrt{(\nu_\Phi + T\nu_\mathcal{F} + \log\delta^{-1})/N_{\boldsymbol{\omega}}}$-neighborhood of $\bar{\Phi}^{\boldsymbol{\omega}}$[6], there exists a specific $\bar{\phi}^{\boldsymbol{\omega}} \in \bar{\Phi}^{\boldsymbol{\omega}}$ such that $\phi$ is $(\rho, C_\rho)$-transferable, then there exists another $C' = C'(L_\ell, L_\mathcal{F}, C_\Phi, C_\mathcal{F}, C_\rho, \rho)$ such that with probability at least $1 - \delta$, we have*

$$\mathcal{L}_0(\widehat{\phi}, \widehat{f}_0) - \mathcal{L}_0(\phi_0^\star, f_0^\star) \leq C'\left[\left(\frac{\nu_\mathcal{F} + \log(1/\delta)}{n_0}\right)^{\frac{1}{2}} + \left(\frac{\nu_\Phi + T\nu_\mathcal{F} + \log(1/\delta)}{N_{\boldsymbol{\omega}}}\right)^{\frac{1}{2\rho}}\right]$$

$$+ \texttt{dist}\left(\sum_{t=1}^T \omega_t \mathcal{D}_t, \mathcal{D}_0\right). \tag{A.1}$$

*Proof.* See Appendix A.3. $\qquad\square$

Note that the task distance naturally appears in the upper bound above. This theorem can be regarded as a predecessor of Theorem 3.1, and we refer readers to Section 3.2 for a detailed exposition on the meaning of each term in the upper bound.

### A.2   A SUFFICIENT CONDITION FOR EXACTLY MATCHING $\bar{\phi}^{\boldsymbol{\omega}} = \phi_0^\star$

**Proposition A.1** (Weighting and task distance minimization). *Suppose that for every $\phi \in \Phi$, there exists two source tasks $1 \leq t_1, t_2 \leq T$ such that $\mathcal{L}_{t_1}^\star(\phi) \leq \mathcal{L}_0^\star(\phi) \leq \mathcal{L}_{t_2}^\star(\phi)$. Then there exists a choice of weights $\boldsymbol{\omega}$, possibly depending on $\phi$, such that $\texttt{dist}(\sum_{t=1}^T \omega_t \mathcal{D}_t, \mathcal{D}_0) = 0$.*

*Proof.* By construction, it suffices to show the existence of some $\boldsymbol{\omega}$ such that

$$\sum_{t=1}^T \omega_t \mathcal{L}_t^\star(\phi) = \mathcal{L}_0^\star(\phi), \qquad \forall \phi \in \Phi. \tag{A.2}$$

---

[6]Recall that a representation $\phi$ is in an $\varepsilon$-neighborhood of $\bar{\Phi}^{\boldsymbol{\omega}}$ if $\sum_{t=1}^T \omega_t[\mathcal{L}_t^\star(\phi) - \mathcal{L}_t^\star(\bar{\phi}^{\boldsymbol{\omega}})] \leq \varepsilon$.

By assumption, fix any $\phi \in \Phi$, we can find $1 \leq t_1, t_2 \leq T$ such that $\mathcal{L}_{t_1}^\star(\phi) \leq \mathcal{L}_0^\star(\phi) \leq \mathcal{L}_{t_2}^\star(\phi)$. We set $\omega_t = 0$ for any $t \notin \{t_1, t_2\}$. If $\mathcal{L}_{t_1}^\star(\phi) = \mathcal{L}_0^\star(\phi) = \mathcal{L}_{t_2}^\star(\phi)$, then any choice of $\omega_{t_1}, \omega_{t_2}$ will suffice. Otherwise, we let

$$\omega_{t_1} = \frac{\mathcal{L}_{t_2}^\star(\phi) - \mathcal{L}_0^\star(\phi)}{\mathcal{L}_{t_2}^\star(\phi) - \mathcal{L}_{t_1}^\star(\phi)}, \qquad \omega_{t_2} = \frac{\mathcal{L}_0^\star(\phi) - \mathcal{L}_{t_1}^\star(\phi)}{\mathcal{L}_{t_2}^\star(\phi) - \mathcal{L}_{t_1}^\star(\phi)}.$$

It is straightforward to check that such a choice of $\boldsymbol{\omega}$ indeed ensures (A.2). The proof is concluded. $\qquad\square$

## A.3 PROOF OF THEOREM A.1

We start by stating two useful lemmas.

**Lemma A.1** (Error for learning the imperfect representation from source data). *Under the setup of Theorem A.1, there exists a constant $C_1 = C_1(L_\ell, L_\mathcal{F}, C_\Phi, C_\mathcal{F}) > 0$ such that for any $\delta \in (0,1)$, we have*

$$\sum_{t=1}^T \omega_t[\mathcal{L}_t^\star(\widehat{\phi}) - \mathcal{L}_t^\star(\bar{\phi}^{\boldsymbol{\omega}})] \leq C_1 \cdot \sqrt{\frac{\nu_\Phi + T\nu_\mathcal{F} + \log(1/\delta)}{N_{\boldsymbol{\omega}}}} \tag{A.3}$$

*with probability at least $1 - \delta$.*

*Proof.* See Appendix A.3.1. $\qquad\square$

**Lemma A.2** (Error for learning the task-specific function from target data). *Under the setup of Theorem A.1, there exists a constant $C_2 = C_2(L_\ell, C_\mathcal{F}) > 0$ such that for any $\delta \in (0,1)$, we have*

$$\mathcal{L}_0(\widehat{\phi}, \widehat{f}_0) - \mathcal{L}_0^\star(\widehat{\phi}) \leq C_2 \cdot \sqrt{\frac{\nu_\mathcal{F} + \log(1/\delta)}{n_0}} \tag{A.4}$$

*with probability at least $1 - \delta$.*

*Proof.* See Appendix A.3.2. $\qquad\square$

To prove Theorem A.1, we start by writing

$$\mathcal{L}_0(\widehat{\phi}, \widehat{f}_0) - \mathcal{L}_0(\phi_0^\star, f^\star) = \mathcal{L}_0(\widehat{\phi}, \widehat{f}_0) - \mathcal{L}_0^\star(\widehat{\phi}) + \mathcal{L}_0^\star(\widehat{\phi}) - \mathcal{L}_0^\star(\bar{\phi}^{\boldsymbol{\omega}}) + \mathcal{L}_0^\star(\bar{\phi}^{\boldsymbol{\omega}}) - \mathcal{L}_0^\star(\phi_0^\star).$$

Suppose the two high probability events in Lemmas A.1 and A.2 hold. Then we can bound $\mathcal{L}_0(\widehat{\phi}, \widehat{f}_0) - \mathcal{L}_0^\star(\widehat{\phi})$ by (A.4). Meanwhile, since $\widehat{\phi}$ is in a $C\sqrt{(\nu_\Phi + T\nu_\mathcal{F} + \log\delta^{-1})/N_{\boldsymbol{\omega}}}$-neighborhood of $\bar{\Phi}^{\boldsymbol{\omega}}$, we can invoke the transferability assumption to get

$$\mathcal{L}_0^\star(\widehat{\phi}) - \mathcal{L}_0^\star(\bar{\phi}^{\boldsymbol{\omega}}) \leq C_\rho \left( \sum_{t=1}^T \omega_t[\mathcal{L}_t^\star(\phi) - \mathcal{L}_t^\star(\bar{\phi}^{\boldsymbol{\omega}})] \right)^{1/\rho} \leq C_\rho C_1^{1/\rho} \cdot \left( \frac{\nu_\Phi + T\nu_\mathcal{F} + \log(1/\delta)}{N_{\boldsymbol{\omega}}} \right)^{1/2\rho}.$$

Moreover, we have the trivial bound that $\mathcal{L}_0^\star(\bar{\phi}^{\boldsymbol{\omega}}) - \mathcal{L}_0^\star(\phi_0^\star) \leq \sup_{\bar{\phi}^{\boldsymbol{\omega}}} \mathcal{L}_0^\star(\bar{\phi}^{\boldsymbol{\omega}}) - \mathcal{L}_0^\star(\phi_0^\star)$. Assembling the three bounds above gives the desired result.

### A.3.1 PROOF OF LEMMA A.1

We start by writing

$$\sum_{t=1}^T \omega_t[\mathcal{L}_t^\star(\widehat{\phi}) - \mathcal{L}_t^\star(\bar{\phi}^{\boldsymbol{\omega}})]$$

$$\leq \sum_{t=1}^T \omega_t[\mathcal{L}_0(\widehat{\phi}, \widehat{f}_t) - \mathcal{L}_t(\bar{\phi}^{\boldsymbol{\omega}}, \bar{f}_t^{\boldsymbol{\omega}})]$$

$$= \sum_{t=1}^T \omega_t \left( \mathcal{L}_t(\widehat{\phi}, \widehat{f}_t) - \widehat{\mathcal{L}}_t(\widehat{\phi}, \widehat{f}_t) + \widehat{\mathcal{L}}_t(\widehat{\phi}, \widehat{f}_t) - \widehat{\mathcal{L}}_t(\bar{\phi}^{\boldsymbol{\omega}}, \bar{f}_t^{\boldsymbol{\omega}}) + \widehat{\mathcal{L}}_t(\bar{\phi}^{\boldsymbol{\omega}}, \bar{f}_t^{\boldsymbol{\omega}}) - \mathcal{L}_t(\bar{\phi}^{\boldsymbol{\omega}}, \bar{f}_t^{\boldsymbol{\omega}}) \right)$$

$$\leq \sum_{t=1}^{T} \omega_t \bigg( \mathcal{L}_t(\widehat{\phi}, \widehat{f}_t) - \widehat{\mathcal{L}}_t(\widehat{\phi}, \widehat{f}_t) + \widehat{\mathcal{L}}_t(\bar{\phi}^{\boldsymbol{\omega}}, \bar{f}_t^{\boldsymbol{\omega}}) - \mathcal{L}_t(\bar{\phi}^{\boldsymbol{\omega}}, \bar{f}_t^{\boldsymbol{\omega}}) \bigg)$$

$$\leq \sup_{\phi \in \Phi, \{f_t\} \subset \mathcal{F}} \omega_t \bigg( \mathcal{L}_t(\phi, f_t) - \widehat{\mathcal{L}}_t(\phi, f_t) + \widehat{\mathcal{L}}_t(\bar{\phi}^{\boldsymbol{\omega}}, \bar{f}_t^{\boldsymbol{\omega}}) - \mathcal{L}_t(\bar{\phi}^{\boldsymbol{\omega}}, \bar{f}_t^{\boldsymbol{\omega}}) \bigg)$$

$$= \sup_{\phi \in \Phi, \{f_t\} \subset \mathcal{F}} \sum_{t=1}^{T} \omega_t \cdot \frac{1}{n_t} \sum_{i=1}^{n_t} \bigg( \mathcal{L}_t(\phi, f_t) - \ell(f_0 \circ \phi(\boldsymbol{x}_{ti}), y_{ti}) + \ell(\bar{f}^{\boldsymbol{\omega}} \circ \bar{\phi}^{\boldsymbol{\omega}}(\boldsymbol{x}_{ti}), y_{ti}) - \mathcal{L}_t(\bar{\phi}^{\boldsymbol{\omega}}, \bar{f}_t^{\boldsymbol{\omega}}) \bigg),$$

where the first inequality is by $\mathcal{L}_t^{\star}(\cdot) = \min_{f_t \in \mathcal{F}} \mathcal{L}_t(\cdot, f_t)$ and the second inequality is by the fact that $(\widehat{\phi}, \{\widehat{f}_t\}_{t=1}^{T})$ is a minimizer of (2.2). To simplify notations, let $\boldsymbol{z}_{ti} = (\boldsymbol{x}_{ti}, y_{ti})$ and let the right-hand side above be $G(\{\boldsymbol{z}_{ti}\})$. Fix two indices $1 \leq t' \leq T$, $1 \leq i_{t'} \leq n_t$, and let $\{\widetilde{\boldsymbol{z}}_{ti}\}$ be the source datasets formed by replacing $\boldsymbol{z}_{t',i_{t'}}$ with some $\widetilde{\boldsymbol{z}}_{t',i_{t'}} = (\widetilde{\boldsymbol{x}}_{t',i_{t'}}, \widetilde{y}_{t',i_{t'}}) \in \mathcal{X} \times \mathcal{Y}$. Since $\{\boldsymbol{z}_{ti}\}$ and $\{\widetilde{\boldsymbol{z}}_{ti}\}$ differ by only one example, we have

$$\sum_{t \neq t'} \sum_{i=1}^{n_t} \frac{\omega_t}{n_t} \bigg( \mathcal{L}_t(\phi, f_t) - \ell(f_t \circ \phi(\boldsymbol{x}_{ti}), y_{ti}) + \ell(\bar{f}_t^{\boldsymbol{\omega}} \circ \phi^{\boldsymbol{\omega}}(\boldsymbol{x}_{ti}), y_{ti}) + \mathcal{L}_t(\bar{\phi}^{\boldsymbol{\omega}}, \bar{f}_t^{\boldsymbol{\omega}}) \bigg)$$

$$+ \sum_{i \neq i_{t'}} \frac{\omega_{t'}}{n_{t'}} \bigg( \mathcal{L}_{t'}(\phi, f_{t'}) - \ell(f_{t'} \circ \phi(\boldsymbol{x}_{t',i}), y_{t',i}) + \ell(\bar{f}_{t'}^{\boldsymbol{\omega}} \circ \phi^{\boldsymbol{\omega}}(\boldsymbol{x}_{t',i}), y_{t',i}) + \mathcal{L}_{t'}(\bar{\phi}^{\boldsymbol{\omega}}, \bar{f}_{t'}^{\boldsymbol{\omega}}) \bigg)$$

$$+ \frac{\omega_{t'}}{n_{t'}} \bigg( \mathcal{L}_{t'}(\phi, f_{t'}) - \ell(f_{t'} \circ \phi(\boldsymbol{x}_{t',i_{t'}}), y_{t',i_{t'}}) + \ell(\bar{f}_{t'}^{\boldsymbol{\omega}} \circ \phi^{\boldsymbol{\omega}}(\boldsymbol{x}_{t',i_{t'}}), y_{t',i_{t'}}) + \mathcal{L}_{t'}(\bar{\phi}^{\boldsymbol{\omega}}, \bar{f}_{t'}^{\boldsymbol{\omega}}) \bigg)$$

$$- G(\{\widetilde{\boldsymbol{z}}_{ti}\})$$

$$\leq \frac{\omega_{t'}}{n_{t'}} \bigg[ \bigg( \mathcal{L}_{t'}(\phi, f_{t'}) - \ell(f_{t'} \circ \phi(\boldsymbol{x}_{t',i_{i'}}), y_{t',i_{t'}}) \bigg) - \bigg( \mathcal{L}_{t'}(\bar{\phi}^{\boldsymbol{\omega}}, \bar{f}_{t'}^{\boldsymbol{\omega}}) - \ell(\bar{f}_{t'}^{\boldsymbol{\omega}} \circ \bar{\phi}^{\boldsymbol{\omega}}(\boldsymbol{x}_{t',i_{t'}}), y_{t',i_{t'}}) \bigg)$$

$$+ \bigg( \mathcal{L}_{t'}(\phi, f_{t'}) - \ell(f_{t'} \circ \phi(\widetilde{\boldsymbol{x}}_{t',i_{i'}}), \widetilde{y}_{t',i_{t'}}) \bigg) - \bigg( \mathcal{L}_{t'}(\bar{\phi}^{\boldsymbol{\omega}}, \bar{f}_{t'}^{\boldsymbol{\omega}}) - \ell(\bar{f}_{t'}^{\boldsymbol{\omega}} \circ \bar{\phi}^{\boldsymbol{\omega}}(\widetilde{\boldsymbol{x}}_{t',i_{t'}}), \widetilde{y}_{t',i_{t'}}) \bigg) \bigg]$$

$$\leq \frac{4\omega_{t'}}{n_{t'}}, \tag{A.5}$$

where the last inequality is by the fact that the loss function is bounded in $[0, 1]$. Taking the supremum over $\phi \in \Phi, \{f_t\} \subset \mathcal{F}$ at both sides, we get $G(\{\boldsymbol{z}_{ti}\}) - G(\{\widetilde{\boldsymbol{z}}_{ti}\}) \leq 4\omega_{t'}/n_{t'}$. A symmetric argument shows that the reverse inequality, namely $G(\{\widetilde{\boldsymbol{z}}_{ti}\}) - G(\{\boldsymbol{z}_{ti}\}) \leq 4\omega_{t'}/n_{t'}$, is also true. That is, we have shown

$$|G(\{\boldsymbol{z}_{ti}\}) - G(\{\widetilde{\boldsymbol{z}}_{ti}\})| \leq \frac{4\omega_{t'}}{n_{t'}}.$$

This means that we can invoke McDiarmid's inequality to get

$$\mathbb{P}\bigg( G(\{\boldsymbol{z}_{ti}\}) - \mathbb{E}[G(\{\boldsymbol{z}_{ti}\})] \geq \varepsilon \bigg) \leq \exp\bigg\{ \frac{-2\varepsilon^2}{\sum_{t=1}^{T} \sum_{i=1}^{n_t} 16\omega_t^2/n_t^2} \bigg\}$$

for any $\varepsilon > 0$, or equivalently

$$G(\{\boldsymbol{z}_{ti}\}) \leq \mathbb{E}[G(\{\boldsymbol{z}_{ti}\})] + 2\sqrt{2} \cdot \sqrt{\frac{\log(1/\delta)}{N_{\boldsymbol{\omega}}}} \tag{A.6}$$

with probability at least $1 - \delta$ for any $\delta \in (0, 1)$. To bound the expectation term, we use a standard symmetrization argument (see, e.g., Lemma 11.4 in (Boucheron et al., 2013)) to get

$$\sqrt{N_{\boldsymbol{\omega}}}\mathbb{E}[G(\{\boldsymbol{z}_{ti}\})] \leq 2\sqrt{N_{\boldsymbol{\omega}}}\mathbb{E}\bigg[ \sup_{\phi \in \Phi, \{f_t\} \subset \mathcal{F}} \sum_{t=1}^{T} \sum_{i=1}^{n_t} \varepsilon_{ti} \cdot \frac{\omega_t}{n_t} \bigg( -\ell(f_t \circ \phi(\boldsymbol{x}_{ti}), y_{ti}) + \ell(\bar{f}_t \circ \bar{\phi}(\boldsymbol{x}_{ti}), y_{ti}) \bigg) \bigg],$$

where the expectation is taken over the randomness in both the source datasets $\{\boldsymbol{z}_{ti}\}$ and the i.i.d. symmetric Rademacher random variables $\{\varepsilon_{ti}\}$. Consider the function space $\mathscr{G} := \{(\phi, \{f_t\}_{t=1}^{T}) : \phi \in \Phi, \{f_t\} \subset \mathcal{F}\}$. Let

$$M_g := \sqrt{N_{\boldsymbol{\omega}}} \cdot \sum_{t=1}^{T} \sum_{i=1}^{n_t} \varepsilon_{ti} \cdot \frac{\omega_t}{n_t} \cdot [-\ell(f_t \circ \phi(\boldsymbol{x}_{ti}), y_{ti})], \qquad g = (\phi, \{f_t\}_{t=1}^{T}) \in \mathscr{G}$$

be the empirical process indexed by the function space $\mathscr{G}$. Conditional on the randomness in the data $\{z_{ti}\}$, this is a Rademacher process with sub-Gaussian increments:

$$\log \mathbb{E} e^{\lambda(M_g - M_{\widetilde{g}})} \leq \frac{\lambda^2}{2} \mathsf{d}^2(g, \widetilde{g}), \qquad \forall \lambda \geq 0, g = (\phi, \{f_t\}_{t=1}^T), \widetilde{g} = (\widetilde{\phi}, \{\widetilde{f}_t\}_{t=1}^T) \in \mathscr{G},$$

where the pseudometric

$$\mathsf{d}^2(g, \widetilde{g}) := N_{\boldsymbol{\omega}} \cdot \sum_{t=1}^T \sum_{i=1}^{n_t} \frac{\omega_t^2}{n_t^2} \bigg( \ell(f_t \circ \phi(\boldsymbol{x}_{ti}), y_{ti}) - \ell(\widetilde{f}_t \circ \widetilde{\phi}(\boldsymbol{x}_{ti}), y_{ti}) \bigg)^2 \leq N_{\boldsymbol{\omega}} \cdot \sum_{t=1}^T \frac{\omega_t^2}{n_t} = 1.$$

Thus, we can invoke Dudley's entropy integral inequality (see, e.g., Corollary 13.2 in (Boucheron et al., 2013)) to get

$$\mathbb{E}[\sup_{g \in \mathscr{G}} M_g - M_{\bar{g}} \mid \{z_{ti}\}] \lesssim \int_0^1 \sqrt{\log \mathcal{N}(\mathscr{G}; \mathsf{d}; \varepsilon)} d\varepsilon,$$

where $\bar{g} = (\bar{\phi}^{\boldsymbol{\omega}}, \{\bar{f}_t^{\boldsymbol{\omega}}\})$, and $\mathcal{N}(\mathscr{G}; \mathsf{d}; \varepsilon)$ is the $\varepsilon$-covering number of $\mathscr{G}$ with respect to the pseudometric d. Taking expectation over the randomness in $\{z_{ti}\}$, we get

$$\mathbb{E}[G(\{z_{ti}\})] \lesssim N_{\boldsymbol{\omega}}^{-1/2} \cdot \int_0^1 \sqrt{\log \mathcal{N}(\mathscr{G}; \mathsf{d}; \varepsilon)} d\varepsilon. \tag{A.7}$$

We now bound the covering number of $\mathscr{G}$. To do so, we define

$$\mathbb{Q} := \sum_{t=1}^T N_{\boldsymbol{\omega}} \cdot \frac{\omega_t^2}{n_t} \cdot \sum_{i=1}^{n_t} \frac{\delta_{\boldsymbol{x}_{ti}}}{n_t}, \qquad \mathbb{Q}_t := \frac{1}{n_t} \sum_{i=1}^{n_t} \delta_{\boldsymbol{x}_{ti}},$$

where $\delta_{\boldsymbol{x}_{ti}}$ is a point mass at $\boldsymbol{x}_{ti}$. Let $\{\phi^{(1)}, \dots, \phi^{(N_\varepsilon)}\} \subset \Phi$ be an $\varepsilon$-covering of $\Phi$ with respect to $L^2(\mathbb{Q})$, where $N_\varepsilon = \mathcal{N}(\Phi; L^2(\mathbb{Q}); \varepsilon)$. This means that for any $\phi \in \Phi$, there exists $j \in \{1, \dots, N_\varepsilon\}$ such that

$$\|\phi - \phi^{(j)}\|_{L^2(\mathbb{Q})}^2 := \sum_{t=1}^T \sum_{i=1}^n N_{\boldsymbol{\omega}} \cdot \frac{\omega_t^2}{n_t^2} \|\phi(\boldsymbol{x}_{ti}) - \phi^{(j)}(\boldsymbol{x}_{ti})\|^2 \leq \varepsilon^2.$$

Now for each $j \in \{1, \dots, N_\varepsilon\}$ and $t \in \{1, \dots, T\}$, let $\{f_t^{(j,1)}, \dots, f_t^{(j, N_\varepsilon^{(j)})}\}$ be an $\varepsilon$-covering of $\mathcal{F}$ with respect to $L^2(\phi^{(j)} \# \mathbb{Q}_t)$, where $\phi^{(j)} \# \mathbb{Q}_t$ is the pushforward of $\mathbb{Q}_t$ by $\phi^{(j)}$, and $N_\varepsilon^{(j)} = \mathcal{N}(\mathcal{F}; L^2(\phi^{(j)} \# \mathbb{Q}_t); \varepsilon)$ has no dependence on $t$ due to the uniform entropy control from Assumption B. This means that for any $f_t \in \mathcal{F}, j \in \{1, \dots, N_\varepsilon\}$, there exists $k \in \{1, \dots, N_\varepsilon^{(j)}\}$ such that

$$\|f_t - f_t^{(j,k)}\|_{L^2(\phi^{(j)} \# \mathbb{Q}_t)}^2 := \frac{1}{n_t} \sum_{i=1}^{n_t} \bigg( f_t \circ \phi^{(j)}(\boldsymbol{x}_{ti}) - f_t^{(j,k)} \circ \phi^{(j)}(\boldsymbol{x}_{ti}) \bigg)^2 \leq \varepsilon^2.$$

Now, let us fix $(\phi, \{f_t\}_{t=1}^T) \in \mathscr{G}$. By construction, we can find $j \in \{1, \dots, N_\varepsilon\}$ and $k_t \in \{1, \dots, N_\varepsilon^{(j)}\}$ for any $1 \leq t \leq T$ such that

$$\|\phi - \phi^{(j)}\|_{L^2(\mathbb{Q})} \leq \varepsilon, \qquad \|f_t - f_t^{(j,k_t)}\|_{L^2(\phi^{(j)} \# \mathbb{Q}_t)} \leq \varepsilon, \ \ \forall 1 \leq t \leq T. \tag{A.8}$$

Thus, we have

$$\mathsf{d}^2 \bigg( (\phi, \{f_t\}_{t=1}^T), (\phi^{(j)}, \{f_t^{(j,k_t)}\}_{t=1}^T) \bigg)$$

$$= N_{\boldsymbol{\omega}} \cdot \sum_{t=1}^T \sum_{i=1}^{n_t} \frac{\omega_t^2}{n_t^2} \bigg( \ell(f_t \circ \phi(\boldsymbol{x}_{ti}), y_{ti}) - \ell(f_t^{(j,k_t)} \circ \phi^{(j)}(\boldsymbol{x}_{ti}), y_{ti}) \bigg)^2$$

$$\leq L_\ell^2 N_{\boldsymbol{\omega}} \cdot \sum_{t=1}^T \sum_{i=1}^{n_t} \frac{\omega_t^2}{n_t^2} \bigg( f_t \circ \phi(\boldsymbol{x}_{ti}) - f_t^{(j,k_t)} \circ \phi^{(j)}(\boldsymbol{x}_{ti}) \bigg)^2$$

$$= L_\ell^2 N_{\boldsymbol{\omega}} \cdot \sum_{t=1}^T \sum_{i=1}^{n_t} \frac{\omega_t^2}{n_t^2} \bigg( f_t \circ \phi(\boldsymbol{x}_{ti}) - f_t \circ \phi^{(j)}(\boldsymbol{x}_{ti}) + f_t \circ \phi^{(j)}(\boldsymbol{x}_{ti}) - f_t^{(j,k_t)} \circ \phi^{(j)}(\boldsymbol{x}_{ti}) \bigg)^2$$

$$\leq 2L_\ell^2 N_{\boldsymbol{\omega}} \sum_{t=1}^{T} \sum_{i=1}^{n_t} \frac{\omega_t^2}{n_t^2} \cdot L_{\mathcal{F}}^2 \|\phi(\boldsymbol{x}_{ti}) - \phi^{(j)}(\boldsymbol{x}_{ti})\|^2 + 2L_\ell^2 N_{\boldsymbol{\omega}} \sum_{t=1}^{T} \frac{\omega_t^2}{n_t} \cdot \frac{1}{n_t} \sum_{i=1}^{n_t} \left( f_t \circ \phi^{(j)}(\boldsymbol{x}_{ti}) - f_t^{(j,k_t)} \circ \phi^{(j)}(\boldsymbol{x}_{ti}) \right)^2$$

$$= 2L_\ell^2 L_{\mathcal{F}}^2 \|\phi - \phi^{(j)}\|_{L^2(\mathbb{Q})}^2 + 2L_\ell^2 N_{\boldsymbol{\omega}} \sum_{t=1}^{T} \frac{\omega_t^2}{n_t} \cdot \|f_t - f_t^{(j,k_t)}\|_{L^2(\phi^{(j)} \# \mathbb{Q}_t)}^2$$

$$\leq 2L_\ell^2 (L_{\mathcal{F}}^2 + 1)\varepsilon^2.$$

This yields

$$\mathcal{N}(\mathscr{G}; \mathsf{d}; \sqrt{2}L_\ell\sqrt{L_{\mathcal{F}}^2 + 1} \cdot \varepsilon) \leq \left| \left\{ (\phi^{(j)}, \{f_t^{(j,k_t)}\}_{t=1}^T) : 1 \leq j \leq N_\varepsilon, 1 \leq k_t \leq N_\varepsilon^{(j)}, \forall 1 \leq t \leq T \right\} \right|$$

$$= N_\varepsilon \cdot (N_\varepsilon^{(j)})^T$$

$$\leq \left( \frac{C_\Phi}{\varepsilon} \right)^{\nu_\Phi} \cdot \left( \frac{C_{\mathcal{F}}}{\varepsilon} \right)^{T\nu_{\mathcal{F}}},$$

from which we get

$$\log \mathcal{N}(\mathscr{G}; \mathsf{d}; \varepsilon) \leq \nu_\Phi \log(C_\Phi L_\ell \sqrt{2(L_{\mathcal{F}}^2 + 1)}) + T\nu_{\mathcal{F}} \log(C_{\mathcal{F}} L_\ell \sqrt{2(L_{\mathcal{F}}^2 + 1)}) + (\nu_\Phi + T\nu_{\mathcal{F}}) \log(1/\varepsilon)$$

$$\lesssim (\nu_\Phi + T\nu_{\mathcal{F}})(1 + \log(1/\varepsilon)).$$

Plugging the above inequality to (A.7), we get

$$\mathbb{E}[G(\{\boldsymbol{z}_{ti}\})] \lesssim N_{\boldsymbol{\omega}}^{-1/2} \cdot \sqrt{\nu_\Phi + T\nu_{\mathcal{F}}} \cdot \left( 1 + \int_0^1 \sqrt{\log(1/\varepsilon)} d\varepsilon \right) \lesssim \sqrt{\frac{\nu_\Phi + T\nu_{\mathcal{F}}}{N_{\boldsymbol{\omega}}}}.$$

The proof is concluded by plugging the above inequality to (A.6).

### A.3.2 PROOF OF LEMMA A.2

Since $\widehat{\phi}$ is obtained from the source datasets $\{S_t\}_{t=1}^T$, it is independent of the target data $S_0$. Throughout the proof, we condition on the randomness in the source datasets, thus effectively treating $\widehat{\phi}$ as fixed. Let $f_{0,\widehat{\phi}} \in \operatorname{argmin}_{f_0 \in \mathcal{F}} \mathcal{L}_0(\widehat{\phi}, f)$. We start by writing

$$\mathcal{L}_0(\widehat{\phi}, \widehat{f}_0) - \mathcal{L}_0^\star(\widehat{\phi}) = \mathcal{L}_0(\widehat{\phi}, \widehat{f}_0) - \widehat{\mathcal{L}}_0(\widehat{\phi}, \widehat{f}_0) + \widehat{\mathcal{L}}_0(\widehat{\phi}, \widehat{f}_0) - \widehat{\mathcal{L}}_0(\widehat{\phi}, f_{0,\widehat{\phi}}) + \widehat{\mathcal{L}}_0(\widehat{\phi}, f_{0,\widehat{\phi}}) - \mathcal{L}_0(\widehat{\phi}, f_{0,\widehat{\phi}})$$

$$\leq \mathcal{L}_0(\widehat{\phi}, \widehat{f}_0) - \widehat{\mathcal{L}}_0(\widehat{\phi}, \widehat{f}_0) + \widehat{\mathcal{L}}_0(\widehat{\phi}, f_{0,\widehat{\phi}}) - \mathcal{L}_0(\widehat{\phi}, f_{0,\widehat{\phi}})$$

$$\leq \sup_{f_0 \in \mathcal{F}} \mathcal{L}_0(\widehat{\phi}, f_0) - \widehat{\mathcal{L}}(\widehat{\phi}, f_0) + \widehat{\mathcal{L}}_0(\widehat{\phi}, f_{0,\widehat{\phi}}) - \mathcal{L}_0(\widehat{\phi}, f_{0,\widehat{\phi}}).$$

The right-hand side above is an empirical process indexed by $f_0 \in \mathcal{F}$. Using similar arguments as those appeared in the proof of Lemma A.1, we have

$$\sup_{f_0 \in \mathcal{F}} \mathcal{L}_0(\widehat{\phi}, f_0) - \widehat{\mathcal{L}}(\widehat{\phi}, f_0) + \widehat{\mathcal{L}}_0(\widehat{\phi}, f_{0,\widehat{\phi}}) - \mathcal{L}_0(\widehat{\phi}, f_{0,\widehat{\phi}}) \lesssim \sqrt{\frac{\nu_{\mathcal{F}} + \log(1/\delta)}{n_0}}$$

with probability at least $1 - \delta$, which is exactly the desired result.

### A.4 PROOF OF THEOREM 3.1

The proof bears similarities to the proof of Theorem A.1, with additional complications in ensuring a uniform control over the learned weights. Let $f_{0,\widehat{\phi}} \in \operatorname{argmin}_{f_0 \in \mathcal{F}} \mathcal{L}_0(\widehat{\phi}, f_0)$. Note that

$$\mathcal{L}_0(\widehat{\phi}, \widehat{f}_0) - \mathcal{L}_0(\phi_0^\star, f_0^\star)$$
$$= \mathcal{L}_0(\widehat{\phi}, \widehat{f}_0) - \widehat{\mathcal{L}}_0^{(2)}(\widehat{\phi}, \widehat{f}_0) + \widehat{\mathcal{L}}_0^{(2)}(\widehat{\phi}, \widehat{f}_0) - \widehat{\mathcal{L}}_0^{(2)}(\widehat{\phi}, f_{0,\widehat{\phi}})$$
$$\quad + \widehat{\mathcal{L}}_0^{(2)}(\widehat{\phi}, f_{0,\widehat{\phi}}) - \mathcal{L}_0(\widehat{\phi}, f_{0,\widehat{\phi}}) + \mathcal{L}_0^\star(\widehat{\phi}) - \mathcal{L}_0^\star(\bar{\phi}^{\widehat{\boldsymbol{\omega}}}) + \mathcal{L}_0^\star(\bar{\phi}^{\widehat{\boldsymbol{\omega}}}) - \mathcal{L}_0^\star(\phi_0^\star)$$

$$\leq \mathcal{L}_0(\widehat{\phi}, \widehat{f}_0) - \widehat{\mathcal{L}}_0^{(2)}(\widehat{\phi}, \widehat{f}_0) + \widehat{\mathcal{L}}_0^{(2)}(\widehat{\phi}, f_{0,\widehat{\phi}}) - \mathcal{L}_0(\widehat{\phi}, f_{0,\widehat{\phi}}) + \mathcal{L}_0^{\star}(\widehat{\phi}) - \mathcal{L}_0^{\star}(\bar{\phi}^{\widehat{\boldsymbol{\omega}}}) + \mathcal{L}_0^{\star}(\bar{\phi}^{\widehat{\boldsymbol{\omega}}}) - \mathcal{L}_0^{\star}(\phi_0^{\star}). \tag{A.9}$$

Let $\widehat{f}_t \in \operatorname{argmin}_{f_t \in \mathcal{F}} \widehat{\mathcal{L}}_t(\widehat{\phi}, f_t)$. We then have

$$\sum_{t=1}^{T} \widehat{\omega}_t [\mathcal{L}_t^{\star}(\widehat{\phi}) - \mathcal{L}_t^{\star}(\bar{\phi}^{\widehat{\boldsymbol{\omega}}})]$$

$$\leq \sum_{t=1}^{T} \widehat{\omega}_t [\mathcal{L}_t(\widehat{\phi}, \widehat{f}_t) - \mathcal{L}_t(\bar{\phi}^{\widehat{\boldsymbol{\omega}}}, \bar{f}_t^{\widehat{\boldsymbol{\omega}}})]$$

$$= \sum_{t=1}^{T} \widehat{\omega}_t \left( \mathcal{L}_t(\widehat{\phi}, \widehat{f}_t) - \widehat{\mathcal{L}}_t(\widehat{\phi}, \widehat{f}_t) + \widehat{\mathcal{L}}_t(\widehat{\phi}, \widehat{f}_t) - \widehat{\mathcal{L}}_t(\bar{\phi}^{\widehat{\boldsymbol{\omega}}}, \bar{f}_t^{\widehat{\boldsymbol{\omega}}}) + \widehat{\mathcal{L}}_t(\bar{\phi}^{\widehat{\boldsymbol{\omega}}}, \bar{f}_t^{\widehat{\boldsymbol{\omega}}}) - \mathcal{L}_t(\bar{\phi}^{\widehat{\boldsymbol{\omega}}}, \bar{f}_t^{\widehat{\boldsymbol{\omega}}}) \right)$$

$$\leq \sum_{t=1}^{T} \widehat{\omega}_t \left( \mathcal{L}_t(\widehat{\phi}, \widehat{f}_t) - \widehat{\mathcal{L}}_t(\widehat{\phi}, \widehat{f}_t) + \widehat{\mathcal{L}}_t(\bar{\phi}^{\widehat{\boldsymbol{\omega}}}, \bar{f}_t^{\widehat{\boldsymbol{\omega}}}) - \mathcal{L}_t(\bar{\phi}^{\widehat{\boldsymbol{\omega}}}, \bar{f}_t^{\widehat{\boldsymbol{\omega}}}) \right)$$

$$\leq 2 \sup_{\phi \in \Phi, \{f_t\} \subset \mathcal{F}, \boldsymbol{\omega} \in \mathcal{W}_\beta} \sum_{t=1}^{T} \omega_t \left( \mathcal{L}_t(\phi, f_t) - \widehat{\mathcal{L}}_t(\phi, f_t) + \widehat{\mathcal{L}}_t(\bar{\phi}^{\boldsymbol{\omega}}, \bar{f}_t^{\boldsymbol{\omega}}) - \mathcal{L}_t(\bar{\phi}^{\boldsymbol{\omega}}, \bar{f}_t^{\boldsymbol{\omega}}) \right).$$

Let $\boldsymbol{z}_{ti} = (\boldsymbol{x}_{ti}, y_{ti})$, and let $\{\widetilde{\boldsymbol{z}}_{ti}\}$ be the source datasets formed by replacing $\boldsymbol{z}_{t',i_{t'}}$ with $\widetilde{\boldsymbol{z}}_{t',i_{t'}}$. Let

$$G(\{\boldsymbol{z}_{ti}\}) := \sup_{\phi \in \Phi, \{f_t\} \subset \mathcal{F}, \boldsymbol{\omega} \in \mathcal{W}_\beta} \sum_{t=1}^{T} \omega_t \left( \mathcal{L}_t(\phi, f_t) - \widehat{\mathcal{L}}_t(\phi, f_t) + \widehat{\mathcal{L}}_t(\bar{\phi}^{\boldsymbol{\omega}}, \bar{f}_t^{\boldsymbol{\omega}}) - \mathcal{L}_t(\bar{\phi}^{\boldsymbol{\omega}}, \bar{f}_t^{\boldsymbol{\omega}}) \right).$$

Then conducting a similar calculation to what led to (A.5), we get

$$|G(\{\boldsymbol{z}_{ti}\}) - G(\{\widetilde{\boldsymbol{z}}_{ti}\})| \leq \frac{4\omega_{t'}}{n_{t'}} \lesssim \frac{\beta}{nT},$$

where the last inequality is by the fact that $\boldsymbol{\omega} \in \mathcal{W}_\beta$ implies $\omega_t \leq \beta/T$ for any $1 \leq t \leq T$. Now, invoking McDiarmid's inequality, we get

$$G(\{\boldsymbol{z}_{ti}\}) \leq \mathbb{E}G(\{\boldsymbol{z}_{ti}\}) + \mathcal{O}\left( \beta \sqrt{\frac{\log(1/\delta)}{nT}} \right) \tag{A.10}$$

with probability at least $1 - \delta$. Now, a standard symmetrization argument plus an application of Dudley's entropy integral bound (similar to what led to (A.7)) gives

$$\mathbb{E}[G(\{\boldsymbol{z}_{ti}\})] \lesssim (nT/\beta^2)^{-1/2} \cdot \int_0^1 \sqrt{\log \mathcal{N}(\mathscr{G}; \mathsf{d}; \varepsilon)} d\varepsilon, \tag{A.11}$$

where now $\mathscr{G} := \{(\phi, \{f_t\}_{t=1}^{T}, \boldsymbol{\omega}) : \phi \in \Phi, \{f_t\} \subset \mathcal{F}, \boldsymbol{\omega} \in \mathcal{W}_\beta\}$, and

$$\mathsf{d}^2 \left( (\phi, \{f_t\}_{t=1}^{T}, \boldsymbol{\omega}), (\widetilde{\phi}, \{\widetilde{f}_t\}_{t=1}^{T}, \widetilde{\boldsymbol{\omega}}) \right)$$

$$\asymp \frac{nT}{\beta^2} \sum_{t=1}^{T} \sum_{i=1}^{n_t} \frac{1}{n_t^2} \left( \omega_t \ell(f_t \circ \phi(\boldsymbol{x}_{ti}), y_{ti}) - \widetilde{\omega}_t \ell(\widetilde{f}_t \circ \widetilde{\phi}(\boldsymbol{x}_{ti}), y_{ti}) \right)^2$$

$$= \frac{nT}{\beta^2} \sum_{t=1}^{T} \sum_{i=1}^{n_t} \frac{\omega_t^2}{n_t^2} \left( \ell(f_t \circ \phi(\boldsymbol{x}_{ti}), y_{ti}) - \ell(\widetilde{f}_t \circ \widetilde{\phi}(\boldsymbol{x}_{ti})) \right)^2 + \frac{nT}{\beta^2} \sum_{t=1}^{T} \sum_{i=1}^{n_t} \frac{1}{n_t^2} (\omega_t - \widetilde{\omega}_t)^2 [\ell(\widetilde{f}_t \circ \widetilde{\phi}(\boldsymbol{x}_{ti}), y_{ti})]^2$$

$$\lesssim \frac{1}{nT} \sum_{t=1}^{T} \sum_{i=1}^{n_t} \left( \ell(f_t \circ \phi(\boldsymbol{x}_{ti}), y_{ti}) - \ell(\widetilde{f}_t \circ \widetilde{\phi}(\boldsymbol{x}_{ti})) \right)^2 + T^2 \|\boldsymbol{\omega} - \widetilde{\boldsymbol{\omega}}\|^2,$$

where the last inequality is by $n_t \asymp n, \omega_t \leq \beta/T$ for any $1 \leq t \leq T$ and $\beta \geq 1$. This means that we can construct a $C\varepsilon$-covering of $\mathscr{G}$ by the following two steps (where $C$ is an absolute constant only depending on $L_\ell$ and $L_{\mathcal{F}}$): (1) cover the space $\{(\phi, \{f_t\}_{t=1}^{T})\}$ by the same construction as (A.8);

(2) construct an $\varepsilon/T$-covering of $\mathcal{W}_\beta$ with at most $(cT/\varepsilon)^T$ many points, where $c$ is an absolute constant. Overall, we can construct a $C\varepsilon$ covering $\mathscr{G}$ with $(C_\Phi/\varepsilon)^{\nu_\Phi} \cdot (C_\mathcal{F}/\varepsilon)^{T\nu_\mathcal{F}} \cdot (cT/\varepsilon)^T$ many points. Hence, we have

$$\log \mathcal{N}(\mathscr{G}; \mathsf{d}; \varepsilon) \lesssim (\nu_\Phi + T\nu_\mathcal{F})(\log(1/\varepsilon) + 1) + T(1 + \log T + \log(1/\varepsilon))$$
$$\lesssim (\nu_\Phi + T\nu_\mathcal{F})(\log(1/\varepsilon) + 1) + T \log T,$$

where the last inequality is by $\nu_\mathcal{F} \geq 1$. Plugging this inequality back to (A.10) and (A.11), we get

$$\sum_{t=1}^T \widehat{\omega}_t [\mathcal{L}_t^\star(\widehat{\phi}) - \mathcal{L}_t^\star(\bar{\phi}^{\widehat{\boldsymbol{\omega}}})] \lesssim \beta \sqrt{\frac{\nu_\Phi + \log(1/\delta)}{nT} + \frac{\nu_\mathcal{F} + \log T}{n}}.$$

with probability at least $1 - \delta$. This means that under this high probability event, we can invoke the transferability assumption to conclude the existence of a specific $\bar{\phi}^{\widehat{\boldsymbol{\omega}}}$ with

$$\mathcal{L}_0^\star(\widehat{\phi}) - \mathcal{L}_0^\star(\bar{\phi}^{\widehat{\boldsymbol{\omega}}}) \lesssim C_\rho \beta^{1/\rho} \cdot \left( \frac{\nu_\Phi + \log(1/\delta)}{nT} + \frac{\nu_\mathcal{F} + \log T}{n} \right)^{1/2\rho}.$$

Recalling (A.9), we arrive at

$$\mathcal{L}_0(\widehat{\phi}, \widehat{f}_0) - \mathcal{L}_0(\phi_0^\star, f_0^\star)$$
$$\leq \mathcal{L}_0(\widehat{\phi}, \widehat{f}_0) - \widehat{\mathcal{L}}_0^{(2)}(\widehat{\phi}, \widehat{f}_0) + \widehat{\mathcal{L}}_0^{(2)}(\widehat{\phi}, f_{0,\widehat{\phi}}) - \mathcal{L}_0(\widehat{\phi}, f_{0,\widehat{\phi}}) + \sup_{\bar{\phi}^{\widehat{\boldsymbol{\omega}}}} \mathcal{L}_0^\star(\bar{\phi}^{\widehat{\boldsymbol{\omega}}}) - \mathcal{L}_0^\star(\phi_0^\star)$$

$$+ \mathcal{O}\left( C_\rho \beta^{1/\rho} \cdot \left( \frac{\nu_\Phi + \log(1/\delta)}{nT} + \frac{\nu_\mathcal{F} + \log T}{n} \right)^{1/2\rho} \right)$$

$$\leq \sup_{f_0 \in \mathcal{F}} \left\{ \mathcal{L}_0(\widehat{\phi}, f_0) - \widehat{\mathcal{L}}_0^{(2)}(\widehat{\phi}, f_0) + \widehat{\mathcal{L}}_0^{(2)}(\widehat{\phi}, f_{0,\widehat{\phi}}) - \mathcal{L}_0(\widehat{\phi}, f_{0,\widehat{\phi}}) \right\} + \sup_{\bar{\phi}^{\widehat{\boldsymbol{\omega}}}} \mathcal{L}_0^\star(\bar{\phi}^{\widehat{\boldsymbol{\omega}}}) - \mathcal{L}_0^\star(\phi_0^\star)$$

$$+ \mathcal{O}\left( C_\rho \beta^{1/\rho} \cdot \left( \frac{\nu_\Phi + \log(1/\delta)}{nT} + \frac{\nu_\mathcal{F} + \log T}{n} \right)^{1/2\rho} \right)$$

with probability $1 - \delta$. Since $\widehat{\phi}$ is independent of the second batch of target data $\{(\boldsymbol{x}_{0i}, y_{0i}) : i \in B_2\}$ and $|B_2| \asymp n_0$, a nearly identical argument as that appeared in the proof of Lemma A.2 gives

$$\sup_{f_0 \in \mathcal{F}} \left\{ \mathcal{L}_0(\widehat{\phi}, f_0) - \widehat{\mathcal{L}}^{(2)}(\widehat{\phi}, f_0) + \widehat{\mathcal{L}}_0^{(2)}(\widehat{\phi}, f_{0,\widehat{\phi}}) - \mathcal{L}_0(\widehat{\phi}, f_{0,\widehat{\phi}}) \right\} \lesssim \sqrt{\frac{\nu_\mathcal{F} + \log(1/\delta)}{n_0}}$$

with probability at least $1 - \delta$. We conclude the proof by invoking a union bound.

# B EXPERIMENTAL DETAILS AND ADDITIONAL RESULTS

| Target Task / Learning Paradigm | PoS | Predicate Detection | NER | Avg |
|---|---|---|---|---|
| Single-Task Learning | 85.06 | 71.51 | 54.96 | 70.51 |
| Pre-Training | 88.66 | 78.57 | 58.22 | 75.15 |
| Weighted Pre-Training | **89.31** *** | **79.21** *** | **60.09** *** | **76.20** |
| Joint Training | 87.23 | 74.90 | 59.55 | 73.89 |
| Weighted Joint Training | **90.71** *** | **76.29** *** | **64.49** *** | **77.16** |

Table 2: Compared to Table 1, more training examples are considered here. Here we only consider the three tasks in Ontonotes 5.0, i.e., PoS tagging, predicate detection, and NER. For each setting, we choose one task as the target task and the remaining two tasks as source tasks. We randomly choose $20K$ training sentences for each source task respectively. As for the target task, we randomly choose 500, 600, 1000 training sentences for PoS tagging, predicate detection, and NER respectively.

In this section, we briefly highlight some important settings in our experiments. More details can be found in our released code. It usually costs about half an hour to run the experiment for each setting (e.g. one number in Table 1) on one GeForce RTX 2080 GPU.

| Learning Paradigm ＼ Setting | PoS + **NER** | PoS + Chunking + **NER** | PoS + Chunking + **Predicate Detection** | Avg |
|---|---|---|---|---|
| Single-Task Learning | 68.47 | 68.47 | 78.00 | 71.65 |
| Joint Training | 65.70 | 67.12 | 79.81 | 70.88 |
| Weighted Joint Training | **69.58** | **69.73** | **80.38** | **73.23** |

Table 3: Additional experiments to illustrate the benefits of weighted joint training compared to joint training. For each setting, the task in the bold text is the target task and the remaining tasks are source tasks. The improvement from weighted training in the joint training paradigm under various settings indicates the effectiveness of `TAWT`.

| Learning Paradigm ＼ Ratio | 1 | 2 | 4 | 8 | 16 | 32 | 64 | 128 | Avg |
|---|---|---|---|---|---|---|---|---|---|
| Single-Task Learning | 30.08 | 30.08 | 30.08 | 30.08 | 30.08 | 30.08 | 30.08 | 30.08 | 30.08 |
| Normalized Joint Training | 38.55 | 36.53 | 42.37 | 54.18 | 62.08 | 63.58 | 62.85 | 62.33 | 52.81 |
| Weighted Normalized Joint Training | **42.31** | **42.13** | **51.64** | **59.08** | **63.46** | **64.86** | **64.08** | **63.65** | **56.40** |
| Upper Bound | 62.69 | 70.31 | 75.65 | 79.39 | 81.59 | 83.19 | 84.18 | - | - |

Table 4: The performance for the analysis on the ratio between the training sizes of the source and target datasets in Fig. 1. In this part, we use NER as the target task and source tasks include PoS tagging and predicate detection. We keep the training size of the target task as 500 and change the ratio from 1 to 128 on a log scale. Single-task learning denotes learning only with the small target data, and upper bound denotes learning with the target data that has the same size as the overall size of all datasets in cross-task learning. "-" means that we do not have enough training data for the upper bound in that setting. The sampling strategy[7] for training examples we used for the analysis is a little different from that used in the experiments in Table 1, so that the performance of single-task learning for the same setting can be a little different.

**Data.** In our experiments, we mainly use two widely-used NLP datasets, Ontontes 5.0 (Hovy et al., 2006) and CoNLL-2000 (Tjong Kim Sang & Buchholz, 2000). Ontonotes 5.0 is a large multilingual richly annotated corpus covering a lot of NLP tasks, and we use the corresponding English annotations for three sequence tagging tasks, PoS tagging, predicate detection, and NER. CoNLL-2000 is a shared task for another sequence tagging task, chunking. There are about $116K$ sentences, $16K$ sentences, and $12K$ sentences in the training, development, and test sets for tasks in Ontonotes 5.0. The average sentence length for sentences in Ontonotes 5.0 is about 19. As for CoNLL-2000, there are about $9K$ sentences and $2K$ sentences in the training and test sets. The corresponding average sentence length is about 24.

**Tasks.** We consider four common sequence tagging tasks in NLP, PoS tagging, chunking, predicate detection, and NER. PoS tagging aims to assign a particular part of speech, such as nouns and verbs, for each word in the sentence. Chunking divides a sentence into syntactically related non-overlapping groups of words and assigns them with specific types, such as noun phrases and verb phrases. Predicate detection aims to find the corresponding verbal or nominal predicates for each sentence. NER seeks to locate and classify named entities mentioned in each sentence into pre-defined categories such as person names, organizations, and locations. Based on the above datasets, there are 50 labels in PoS tagging, 23 labels in chunking, 2 labels in predicate detection, and 37 labels in NER. As for the evaluation metric, we use accuracy for PoS tagging, span-level F1 for chunking, token-level F1 for predicate detection, and span-level F1 for NER.

**The model.** We use BERT as our basic model in our main experiments. Specifically, we use the pre-trained case-sensitive BERT-base PyTorch implementation (Wolf et al., 2020). We use the common parameter settings for BERT. Specifically, the max length is 128, the batch size is 32, the epoch number is 4, and the learning rate is $5e^{-5}$. In the BERT model, the task-specific function is the last-layer linear classifier, and the representation model is the remaining part.

---

[7]Specifically, in the main experiments, we first randomly sample $9K$ sentences and then randomly sample a specific number of sentences (e.g. 500 sentences for NER) among the $9K$ sentences for the training set of the target task. In the analysis, we first randomly sample $100K$ sentences and then randomly sample a specific number of sentences (e.g. 500 sentences for NER) among the $100K$ sentences for the training set of the target task. We use a different sampling strategy for training sentences in main experiments and analysis, simply because the largest training size of the data we considered in the two situations is different.

| *Target Size* / *Learning Paradigm* | 500 | 1000 | 2000 | 4000 | 8000 | Avg |
|---|---|---|---|---|---|---|
| Single-Task Learning | 30.08 | 54.26 | 68.05 | 74.77 | 79.20 | 61.27 |
| Normalized Joint Training (Ratio=8) | 54.18 | 64.69 | 71.33 | 76.95 | 80.37 | 69.50 |
| Weighted Normalized Joint Training (Ratio=8) | **59.08** | **66.00** | **72.30** | **76.95** | **80.64** | **70.99** |
| Upper Bound | 79.39 | 81.49 | 83.53 | 84.26 | - | - |
| Normalized Joint Training (Ratio=10) | 58.06 | 65.94 | 71.48 | 77.33 | **80.34** | 70.63 |
| Weighted Normalized Joint Training (Ratio=10) | **60.96** | **67.45** | **72.52** | **77.44** | 79.94 | **71.66** |
| Upper Bound | 79.70 | 82.32 | 83.39 | 84.84 | - | - |

Table 5: The performance for the analysis on the training size of the target dataset in Fig. 1. In this part, we use NER as the target task and source tasks include PoS tagging and predicate detection. We keep the ratio between the training sizes of the source and target datasets as 8 (10) and change the training size of the target dataset from 500 to 8000 on a log scale. Single-task learning denotes learning only with the small target data, and upper bound denotes learning with the target data that has the same size as the overall size of all datasets in cross-task learning. "-" means that we do not have enough training data for the upper bound in that setting.

| *Target Task* / *Learning Paradigm* | PoS | Chunking | Predicate Detection | NER |
|---|---|---|---|---|
| Weighted Pre-Training | (0.68, 0.01, 0.31) | (0.32, 0.30, 0.37) | (0.33, 0.35, 0.32) | (0.92, 0.05, 0.04) |
| Weighted Joint Training | (0.04, 0.04, 0.03) **0.89** | (0.04, 0.04, 0.05) **0.87** | (0.04, 0.04, 0.05) **0.87** | (0.04, 0.04, 0.05) **0.87** |
| Weighted Normalized Joint Training | (0.0005, 0.0004, 0.0004) **0.9986** | (0.0005, 0.0005, 0.0004) **0.9987** | (0.002, 0.002, 0.002) **0.995** | (0.003, 0.003, 0.003) **0.990** |

Table 6: The final learned weights on tasks in the experiments in Table 1. For each setting, the final weights on three source tasks are represented by a three-element tuple. The source tasks are always organized in the following order: PoS tagging, chunking, predicate detection, and NER. For example, the tuple (0.68, 0.01, 0.31) for the target task Pos tagging in the weighted pre-training indicate that the final weights on the three source tasks (chunking, predicate detection, NER) are 0.68, 0.01, and 0.31 respectively. As for TAWT in the joint training, we have extra weight on the target task, which is bold in the second line.

**Cross-task learning paradigms.** In our experiments, we consider two cross-task learning paradigms, pre-training, and joint training. Pre-training first pre-trains the representation part on the source data and then fine-tunes the whole target model on the target data. Joint training uses both source and target data to train the shared representation model and task-specific functions for both source and target tasks at the same time. As for the multi-task learning part in both pre-training and joint training, we adopt the same multi-task learning algorithm as in MT-DNN (Liu et al., 2019).

**Experiments with more target training examples.** Compared to the main experiments in Sec. 4, we further experiment with more training examples in the target data for three tasks in Ontonotes 5.0. Without chunking, we also use more training examples in the source tasks. As shown in Table 2, we can see that TAWT is still beneficial even with more training examples, though the relative improvement of TAWT is smaller compared to that with fewer training examples.

**Additional experiments to illustrate the benefits of TAWT.** In this part, we conduct additional experiments with TAWT on the joint training with fewer source tasks compared to the main experiments in Sec. 4. For simplicity, we make the target task bold to distinguish it from source tasks for each setting. For example, in the setting of PoS + Chunking + **NER**, NER is the target task and the other two tasks are source tasks. For PoS + **NER**, we randomly sample $20K$ training sentences for PoS tagging and $2K$ training sentences for NER. For PoS + Chunking + **NER**, we randomly sample $20K$, $9K$[8] and $2K$ training sentences for PoS tagging, chunking, and NER respectively. As for PoS + Chunking + **Predicate Detection**, we randomly sample $20K$, $9K$, and $2K$ training sentences for PoS tagging, chunking, and predicate detection respectively. As shown in Table 3, we can see that TAWT is still beneficial under these diverse settings, which is a complement for our main experiments in Sec. 4.

---

[8]We choose $9K$ training sentences for chunking because the training size of the chunking dataset is 8936.

| Target Task / Learning Paradigm | PoS | Chunking | Predicate Detection | NER | Avg |
|---|---|---|---|---|---|
| Weighted Pre-Training (fixed weights) | 49.17 | 73.05 | 74.26 | 39.84 | 59.08 |
| Weighted Pre-Training (dynamic weights) | **51.17** | **73.41** | **75.77** | **46.23** | **61.64** |
| Weighted Normalized Joint Training (fixed weights) | **86.93** | 90.12 | 74.90 | **63.60** | 78.89 |
| Weighted Normalized Joint Training (dynamic weights) | 86.07 | **90.62** | **76.67** | 63.44 | **79.20** |

Table 7: Comparison between dynamic weights and fixed final weights. There are four tasks in total, PoS tagging, chunking, predicate detection, and NER. For each setting, we choose one task as the target task and the remaining three tasks are source tasks.

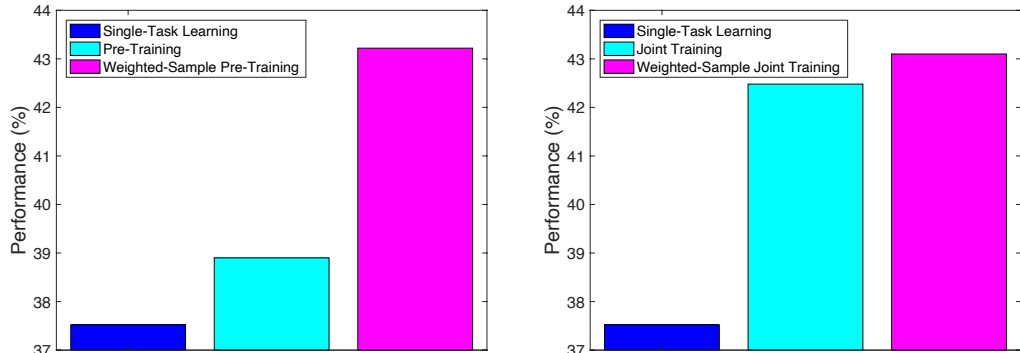

Figure 2: Extension to weighted-sample training. The weighted training algorithm can be easily extended from the weights on tasks to the weights on samples. As for the experiments on weighted-sample training, we use the PoS tagging on entity words as the source task and named entity classification on entity words as the target task. Note that the settings for the weighted-sample training are quite different from those for the weighted-task training in the remaining parts because the weighted-sample training is much more costly compared to the weighted-task training.

**Analysis of some crucial factors.** As shown in Fig. 1, we analyze two crucial factors that affect the improvement of TAWT. First, in general, we find that the improvement of TAWT in normalized joint training is larger when the ratio between the training sizes of the source and target dataset is smaller, though the largest improvement may not be achieved at the smallest ratio. Note that the improvement vanishes with a large ratio mainly because the baseline (normalized joint training) is already good enough with a large ratio. Second, the improvement from TAWT will decrease with the increase of the training size of the target data. In a word, TAWT is more beneficial when the performance of the base model is poorer, either with a smaller target data or with a smaller ratio. The specific performance for experiments in our analysis on two crucial factors for TAWT, the ratio between the training sizes of the source and target datasets, and the training size of the target dataset, can be found in Table 4 and Table 5 respectively.

**Dynamic weights analysis.** As for experiments in Table 1, there are four tasks in total, PoS tagging, chunking, predicate detection, and NER. For each setting, we choose one task as the target task and the remaining three tasks are source tasks. The learned final weights on tasks are shown in Table 6. To better understand our algorithm, we compare TAWT with dynamic weights and TAWT with fixed final weights. TAWT with dynamic weights is the default setting for our algorithm, where weights on tasks for each epoch are different. As for TAWT with fixed weights, we simply initialize the weights as the final weights learned by our algorithm and fix the weights during the training. As shown in Table 7, we find that TAWT with dynamic weights (the default one) is slightly better than TAWT with final fixed weights. It indicates that fixed weights might not be a good choice for weighted training, because the importance of different source tasks may change during the training. In other words, it might be better to choose the weighted training with dynamic weights, where the weights are automatically adjusted based on the state of the trained model.

**Extension to the weighted-sample training.** As shown in Fig. 2, TAWT can be easily extended from the weights on tasks to the weights on samples. We can see that TAWT based on weighted-sample training is also beneficial for both pre-training and joint training. Because the weighted-sample training is much more costly compared to the weighted-task training, we choose a much simpler

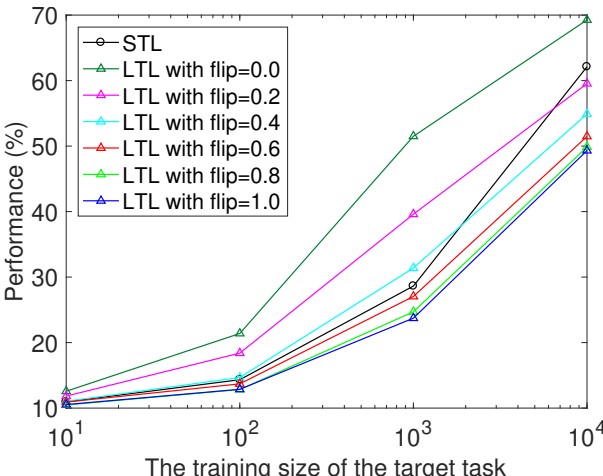

Figure 3: Illustration for the task distance. We find that **the source data is more beneficial when the task distance between the source data and the target data is smaller or the size of the target data is smaller.** In this part, we keep the training size of the source task as 10000 and change the training size of the target task from 10 to 10000 in a log scale. STL denotes the single-task learning only with the target data. LTL denotes the learning to learn paradigm, where we first learn the representations in the source data and then learn the task-specific function in the target data. For the learning to learn paradigm, we consider the source task with different flip rates from 0.0 to 1.0, where the flip rate is an important factor in generating the source data and lower flip rate indicates a smaller task distance between the source data and the target data.

setting here. In this part, we use the PoS tagging on entity words as the source task and named entity classification on entity words as the target task. Note that both tasks we considered here are word-level classification tasks, i.e., predicting the label for a given word in a named entity. We still use the Ontontes 5.0 (Hovy et al., 2006) as our dataset. There are 50 labels in PoS tagging on entity words, and 18 labels [9] in named entity classification on entity words. There are 37534 examples in the development set and 23325 examples in the test set for both source and target tasks. As for the training set, we randomly sample 1000 examples for the source task and 100 examples for the target task. As for the model, we use two-layer NNs with 5-gram features. The two-layer NNs have a hidden size of 4096, ReLU non-linear activation, and cross-entropy loss. As for the embeddings, we use 50 dimensional Glove embeddings (Pennington et al., 2014). The majority baseline for named entity classification on entity words on the test data is $20.17\%$. As for training models, the size of the training batch is 128, the optimizer is Adam (Kingma & Ba, 2015) with a learning rate $3e^{-4}$, and the number of training epochs is 200. As for updating weights on samples, we choose to update the weights every 5 epoch with the mirror descent in Eq. 2.8 and thus 40 updates on weights in total. In this part, we approximate the inverse of the Hessian matrix ($\left[ \sum_{t=1}^{T} \omega_t \nabla_\phi^2 \widehat{\mathcal{L}}_t(\phi^{k+1}, f_t^{k+1}) \right]^{-1}$) in Eq. 2.7 by a constant multiple of the identity matrix as in MAML (Finn et al., 2017), and choose the constant as 5. According to our experiments, the results of this approximation are similar to those of the approximation that we used in Sec. 4. The corresponding results are shown in Fig. 2. In the future, we also plan to group the instances and give each group a weight rather than each sample.

**Settings for simulations on task distance.** In this part, we first randomly generate $10K$ examples $\mathcal{D}$, where the dimension of the inputs are 1000 and the corresponding labels are in $0 - 9$ (10 classes). For each dimension of the input, it is randomly sampled from a uniform distribution over $[-0.5, 0.5]$. For each flip rate $q\% \in [0, 1]$, we randomly choose $q\%$ of the data and replace their labels with a random label uniformly sampled from the 10 classes. The data with flip rate $q\%$ is denoted as $\mathcal{D}_{q\%}$ (including $\mathcal{D}_{0.0}$ for the original dataset). For each dataset $\mathcal{D}_{q\%}$, we train a 2-layer NNs $\mathcal{M}_{q\%}$ with $100\%$ accuracy and almost zero training loss on the dataset. The 2-layer NNs can then be used to generate data for the task $T_{q\%}$. We use the $T_{0.0}$ as the target task and change $q\%$ from 0.0 to 1.0 as

---

[9]In NER, we have 37 labels because each type of entity have two variants of labels (B/I for beside/inside) and one more extra-label $O$ for non-entity words is also considered.

the source task. Based on the process of the data generation, we can see that the flip rate $q\%$ plays an important role in the intrinsic difference between the source task $T_{q\%}$ and the target task $T_{0.0}$. In general, we can expect that a smaller flip rate $q\%$ indicates a smaller task distance between the source task $T_{q\%}$ and the target task $T_{0.0}$. This data generation process is inspired by the teacher-student network in (Hinton et al., 2015). As shown in Fig. 3, we keep the training size of the source task as 10000 and change the training size of the target task from 10 to 10000 in a log scale. For simplicity, we use the same architectures for both the teacher network that we used for generating data and the student network that we used for learning the data, i.e., a two-layer neural network with a hidden size of 4096, an input size of 1000, and an output size of 10. As for training, the size of the training batch is 100, the optimizer is Adam (Kingma & Ba, 2015) with learning rate $3e^{-4}$, and the number of training epochs is 100.

**Details on using existing assets.** As for the corresponding code and pre-trained models for BERT, we directly download it from the Github of huggingface whose license is Apache-2.0, and more details can be found in `https://github.com/huggingface/transformers`. As for the Ontonotes 5.0, we obtain the data from LDC. The corresponding license and other details can be found in `https://catalog.ldc.upenn.edu/LDC2013T19`. As for the CoNLL-2000 shared task, we download the data directly from the website and more details are in `https://www.clips.uantwerpen.be/conll2000/chunking`.

## C    ON THE CHOICE OF EXPERIMENTAL SETTINGS

In this part, we clarify the choice of experimental settings based on the following perspectives.

**Signal selection:** `TAWT` can also be applied to cross-domain or cross-lingual signals. Compared to the above two types of signals, we think cross-task signals are more widespread and more difficult to be used efficiently. Therefore, in our experiments, we choose the most difficult and crucial signals, cross-task signals, to verify the effectiveness of `TAWT`. Another reason is that we didn't find any existing weighted training algorithms with theoretical guarantees for cross-task learning, while weighted training is common for domain adaptation, such as importance sampling.

**The choice of similar domains:** We choose to select tasks with similar domains. Ideally, `TAWT` can be used for cross-task signals in different domains or languages. However, the gap between different domains or different languages can make cross-task learning more complicated. We instead consider cross-task learning in similar domains of English to simplify the settings. But similar domains didn't mean that datasets overlap because we randomly sample sentences for different tasks independently without repetition, causing a small ratio of overlap in our case.

**Area selection:** We choose to experiment with NLP tasks because we think cross-task signals are more common in the NLP area and NLP tasks are more diverse compared to tasks in other areas.

**Dataset selection:** We choose Ontonotes 5.0 as our main dataset because it is widely used, and provides large-scale expert annotations (on 2.9 million words) for a wide range of NLP tasks. This enables us to focus on learning with various tasks in similar domains. As for CoNLL-2000, we add it mainly because we want to analyze the impact of chunking on NER.

**Task selection:** On the one hand, sequence tagging is more challenging than classification tasks. On the other hand, the evaluation of sequence tagging is more reliable, compared with generative tasks. Another reason is that Ontonotes 5.0 mainly cover sequence tagging tasks.

**Model selection:** We choose to use BERT in our main experiments because BERT is widely used and all SOTA models are slight improvements of BERT. For weighted-sample training, we instead choose two-layer NNs with 5-gram features because it is simple and fast. Similar results could be shown even if the model is more complex, but exhaustive experimentation is not our goal.

**The choice of the low-resource setting:** There is an intrinsic trade-off between the base performance of single-task learning and the relevant improvement of cross-task learning. Specifically, if the base performance of single-task learning is good enough, adding cross-task signals can introduce extra noise compared to its information. In our experiments, we simply consider a simple low-resource setting where few target examples are available. Actually, we can still see the effectiveness of `TAWT` in cross-task learning as long as the base performance is not too high, but the relative improvement of `TAWT` will be smaller compared to that in the low-resource setting.

