# OpenReview forum: "Weighted Training for Cross-Task Learning"
_ICLR.cc/2022/Conference — ICLR 2022 Oral_

### Official Review · Reviewer_HcJj · 2021-10-29

**Correctness:** 3
**Technical Novelty And Significance:** 4
**Empirical Novelty And Significance:** 3
**Recommendation:** 8
**Confidence:** 3

**Main Review:**

Review note: As noted in the initial review assessment, I am less comfortable with the mathematics underlying section 3 and Appendix A. I will focus more on the empirical results and I hope other reviewers will help review those. This explains the lower confidence score.

Pros:

- The method is well-motivated. It is clear what the method is aiming for and how it differs from existing approaches. The added theoretical guarantees are welcome.
- The experimental setup is convincing and the results confirm the soundness of the approach. The settings are well described and I appreciated Section B.1. describing some of the rationale behind these choices.
- The writing is very good.
- The appendix is very rich in additional interesting experiments.

Cons:

- Dynamic weights analysis not present in main text:

TAWT is more complex than prior methods. Seeing the importance of dynamic weights throughout training (Table 8) is very relevant as it shows that even a strong choice of initial weights for $\omega$ (e.g through hyperparam opt) may not be as good as TAWT.
Currently, this feels like it is missing from the main body and deserves to be emphasized earlier.

- Weight initialization for joint training
More of a suggestion than a con: For joint training, I believe it would be better to use normalized joint training as the baseline. Indeed, initialization with task weights that depend on the sample size is a stronger baseline and more closely aligned with approaches used in practice.
An added benefit is that table 3 could then be removed (since it’d be the later rows of Table 1 and Table 2). The non normalized results could still be kept in appendix for completeness. This would also allow Table 8 and its discussion to be moved into the main body.

- Minor typos: The writing is good overall, but there are still some minor typos:
e.g: p20 Pre-training first pre-train**s** and then fine-tune**s**


**Summary Of The Paper:**

This paper introduces Target-Aware Weighted Training (TAWT), a cross-task learning algorithm. In the two-step procedure popular for cross-task training, the weight vector $\omega$ defining the relative weight of each source task is usually exogenous (i.e: a hyperparameter). In contrast, in TAWT it is part of the optimization procedure, with the weights depending on the proximity (throughout training) of each source task to the target task (hence the “target aware” name).

The authors first derive an algorithm that enables learning when $ \omega $ is made endogenous in the case of (weighted) pre-training. They then show how it can naturally be extended for weighted joint-training by considering the source tasks as one of the source tasks we learn representations from. They then provide theoretical performance guarantees for TAWT.

The authors apply TAWT on 4 NLP tasks, using BERT as their base model. They evaluate their method on both full-data and limited-data settings on each target task, using the 3 other tasks as source tasks. They show that TAWT-pretraining and TAWT-joint training significantly outperform their unweighted counterparts across tasks. These improvements are more marked when target task data is scarce.

They also show that weight initialization can be particularly important when the different source datasets have different # examples (e.g: when training jointly on data abundant source tasks and the data-scarce target task). This specific finding is not very novel, but it is great to see TAWT-training also helping when initialization weights are chosen correctly.

Critically, the authors also show the importance of varying task weights throughout training. Indeed, fixing the task weights to be the final weights from TAWT leads to worse results than training with TAWT. This suggests that it is important that task-weights vary throughout the training process.


**Summary Of The Review:**

A rare mix of well-motivated method, theoretical guarantees, solid experimental work and convincing results. There are minor presentation improvements to be done but I think this is a strong paper otherwise. As noted earlier, I am not able to review carefully Sec 3/Appendix A so this the caveat to my review.

---

> ### Author Response · Authors · 2021-11-19
> **Response to Reviewer HcJj**
>
> Thank you for your valuable feedback!
>
> **Reply to “Dynamic weights analysis”:** Thanks for your suggestion! We will highlight it in the main text in the revised version.
>
> **Reply to “Normalized joint training”:** Thanks for your suggestion! We will make it more clear in the revised version.  We originally chose to compare with joint training mainly because mainstream models use the joint training instead of normalized joining training, such as BERT [1] and MT-DNN [2].
>
> **Reply to “minor typos”:** Thanks, we will fix them.
>
> [1] Jacob Delvin, Ming-Wei Chang, Kenton Lee, and Kristina Toutanova. "BERT: Pre-training of Deep Bidirectional Transformers for Language Understanding." In NAACL-HLT, 2019.\
> [2] Xiaodong Liu, Pengcheng He, Weizhu Chen, and Jianfeng Gao. "Multi-Task Deep Neural Networks for Natural Language Understanding." In ACL, 2019.

---

### Official Review · Reviewer_jFGp · 2021-11-02

**Correctness:** 3
**Technical Novelty And Significance:** 3
**Empirical Novelty And Significance:** 3
**Recommendation:** 6
**Confidence:** 3

**Main Review:**

Strengths
- The proposed method is novel and well-motived.
- The authors provide a theoretical guarantee of the proposed method.
- Empirical results indicate that the proposed model greatly outperforms the vanilla methods in both pre-training and joint learning. The authors also open-source their implementation.


Weaknesses
- Add a section that introduces "Related Works" would be very helpful for readers to understand the background and the novelty of this work. For example, what're the most popular/advanced methods in cross-task learning? How is this method related to and different from them? What's the relationship between cross-task learning, joint learning, and multi-task learning?
- The empirical evidence would be much more convincing if authors compare their model with other methods besides the vanilla pre-training and joint training. Many existing works that utilize different strategies to calculate the weight of different task has been introduced in the area of multi-task learning. Although this work is designed for "cross-task learning" instead of "multi-task learning", the existing methods can be straightforwardly used to address the same problem. So it would be great if the authors can either compare the proposed methods with existing works or explain why other existing works cannot be directly adapted to this problem. There are many great survey papers that summarize the related works (e.g., [A Comparison of Loss Weighting Strategies for Multi task Learning in Deep Neural Networks](https://ieeexplore.ieee.org/document/8848395) and [A Survey on Multi-Task Learning](https://ieeexplore.ieee.org/stamp/stamp.jsp?arnumber=9392366&tag=1). )

**Summary Of The Paper:**

This paper introduces a novel cross-task training method that weights different tasks by optimizing a representation-based task distance between the source and target tasks. They provide both theoretical guarantee and empirical evidence to corroborate the proposed method.

**Summary Of The Review:**

This is a good paper with novel ideas, but it can be further improved by adding more backgrounds, related works, and comparisons with other state-of-the-art methods.

---

> ### Author Response · Authors · 2021-11-19
> **Response to Reviewer jFGp**
>
> Thank you for your insightful comments!
>
> **Reply to “Related Work”:** Thanks for your suggestion! We will add it in the revised version.
>
> **Reply to “Comparison with other methods”:** Thanks for your suggestion! We will add the comparison in the revised version. We originally chose to compare with vanilla pre-training and joint training for the following two reasons: 1) Our goal is to design a practical algorithm with **theoretical guarantees** instead of simply state-of-the-art (SOTA). In this sense, it’s unfair to compare TAWT with other weighted training methods **without theoretical guarantees**. 2) Current mainstream models in this area still use the vanilla pre-training or joint training, such as BERT [1] and MT-DNN [2].
>
> [1] Jacob Delvin, Ming-Wei Chang, Kenton Lee, and Kristina Toutanova. "BERT: Pre-training of Deep Bidirectional Transformers for Language Understanding." In NAACL-HLT, 2019.\
> [2] Xiaodong Liu, Pengcheng He, Weizhu Chen, and Jianfeng Gao. "Multi-Task Deep Neural Networks for Natural Language Understanding." In ACL, 2019.

---

### Official Review · Reviewer_wLuT · 2021-11-03

**Correctness:** 3
**Technical Novelty And Significance:** 4
**Empirical Novelty And Significance:** 3
**Recommendation:** 8
**Confidence:** 3

**Main Review:**

I enjoyed this paper a great deal. The presentation is very clean and the underlying algorithm and theory is interesting. The results look quite promising for the method.

A few comments:

[1] Are there connections to work that learns data weighting using multi-arm bandits? Specifically:

Automated Curriculum Learning for Neural Networks
Alex Graves, Marc G. Bellemare, Jacob Menick, Remi Munos, Koray Kavukcuoglu

It would be interesting to discuss connections to this work - the ideas are different but there does seem to be some connection.

[2] The choice of mirror descent is not critical, correct? For example projected gradient methods could be used instead? It would be useful to note this.

[3] Given Assumption B, it seems clear that \phi must be a vector, correct? This is a minor thing, but I think that was not specified earlier in the paper?

[4] Definition 3.2 (transferability) could benefit from much more explanation. I’m also not sure if it’s quite correct as written. Should there be a \forall quantifier, for example \forall \phi \in \Phi before Eq. 3.5? Also I’m not sure how this equation implies that the loss is “controlled by a polynomial…”. Finally, is there a reason that the expression (\sum …) is positive? And if it can become negative, how can we take a power to 1/p? Finally, I have very little intuition about what values p will take in practice, or what p intuitively corresponds to.


**Summary Of The Paper:**

The paper discusses an approach that learns to weight data from different tasks in pretraining or mutli-task learning. It also gives a VC/empirical-processes style analysis, which gives guarantees for the algorithm and insights about sample complexity. Finally it describes experiments on a number of NLP problems.


**Summary Of The Review:**

An interesting algorithm, theory, and results. A few parts of the paper need clarification.

---

### Official Review · Reviewer_xsx7 · 2021-11-03

**Correctness:** 3
**Technical Novelty And Significance:** 4
**Empirical Novelty And Significance:** 4
**Recommendation:** 8
**Confidence:** 3

**Main Review:**

The paper proposes a new weighted training algorithm, TAWT, to learn the task-aware weights on tasks for better using the cross-task signals.  The author theoretically and empirically verified the effectiveness of TAWT.
Strength:
1. The paper was clearly written and give a good formal definition for the problem of weighted training.
2. The paper can bring a new research interest for multi-task learning and transfer learning.
3. The experiment designs are very good,which show the affects on different training data size, impact of the ratio between source tasks training samples and target task training samples, etc.

Weakness:
1. The paper introduces a representation-based task distance, but this distance is neglected in the analysis of Eq.(3.8). I think it could not be negligible. Otherwise, the upper bound become task-agnostic.
2. The used four NLP tasks are closely related.  It's better to add an irrelevant task, such as sentiment analysis, to show the effect of weighted training.
3. An illustration of weight vector should be provided.

Others:
1. It's interesting to extend this idea for incorporating some unsupervised tasks, such as Masked language model.


**Summary Of The Paper:**

The paper proposes a new weighted training algorithm, TAWT, to learn the task-aware weights on tasks for better using the cross-task signals.  The author theoretically and empirically verified the effectiveness of TAWT.
The paper can bring a new research interest for multi-task learning and transfer learning.

**Summary Of The Review:**

The paper proposes a new weighted training algorithm, TAWT, to learn the task-aware weights on tasks for better using the cross-task signals.  The proposed method is quite effective for transfer learning and multi-task learning on small target data.
The weighted training is very important but there lacks good work in this direction. Therefore, this paper is great to give an attempt for task-aware weighted training.

---

> ### Author Response · Authors · 2021-11-19
> **Response to Reviewer xsx7**
>
> Thank you for your kind review!
>
> **Reply to “representation-based task distance”:** We would like to point out that the notion of representation-based task distance (Definition 3.1) *appears explicitly* in Equation (3.8). The upper bound in Equation (3.8) is a superposition of three terms, and the last term is precisely the task distance. This task distance is **a function of the weight vector**. Thus, with proper weighting, this term becomes negligible compared to the other two terms, demonstrating the utility of weighted training (See Appendix A.2 for an illustration).
>
> **Reply to “weight vector”:** The illustration of weight vectors can be found in **Table 7** in Appendix B. More discussion on weights can be found in **dynamic weights analysis** in Appendix B. We will make it more clear in the revised version.
>
> **Reply to “additional experiments”:** Thanks for your suggestion on sentiment analysis and masked language model.  We also plan to further evaluate TAWT in more settings.

---

> > ### Comment · Reviewer_xsx7 · 2021-11-29
> > **Thanks for your response.**
> >
> > Thanks for your response.

---

### Decision · Program_Chairs · 2022-01-20

**Decision:**

Accept (Oral)

**Comment:**

The paper proposes an approach to learn the task-specific weights in pretraining or mutli-task learning. It provides theoretical guarantees to the algorithm, as well as strong empirical results on several NLP problems. All the reviewers agreed that the work is interesting and the paper is well written. During the discussion period, the authors committed to address in the revised version (relatively minor) concerns raised by reviewers, including providing additional clarifications and additional comparisons to related methods. Overall, this is a strong paper that merits an acceptance.